# Beyond Behavioral Alignment: Leveraging Core Cognitive Dimensions for Enhanced Human-like MLLMs

## Abstract

The emergence of multimodal large language models (MLLMs) has led to near-human performance across various multimodal cognitive and reasoning tasks, despite relying solely on next-token prediction objectives. A critical and under-explored question is whether MLLMs trained under this paradigm truly exhibit human-like visual conceptual representations and behaviors during multimodal reasoning. To investigate this, we evaluated MLLMs on the widely-used behavioral task of Odd-One-Out (O1O), revealing a limited predictive accuracy for human choices. To address this discrepancy, we propose a novel approach: instead of merely using raw human behavioral data, we first identified core cognitive dimensions and judgmental bases from human behavioral records in O1O experiments. Subsequently, we fine-tuned Qwen2.5-VL in a data-driven manner, guided by these extracted human core cognitive dimensions, thereby markedly enhancing its behavioral consistency with humans. Intriguingly, we found that models aligned with human cognition not only maintain their generality in downstream tasks but can even achieve performance improvements. Furthermore, searchlight representational similarity analysis (RSA) and cortical projection analyses revealed increased activation in brain regions associated with problem planning and decision-making, such as the prefrontal cortex, in the fine-tuned model. This finding potentially offers a neuroscientific explanation for the observed improvements and human-like alignment.

## 1 Introduction

The advent of MLLMs marks a significant milestone in artificial intelligence, demonstrating capabilities that rival human performance across a spectrum of cognitive and reasoning tasks (Bubeck et al., 2023; Team et al., 2023; Bai et al., 2023). These models predicting token by token have developed a remarkable ability to process and integrate information from disparate modalities, such as vision and language. This paradigm has fueled progress in areas from visual question answering to complex multimodal reasoning. However, a fundamental question remains largely underexplored: do these models, in achieving human-level performance, also develop human-like underlying representations and behavioral patterns? Simply matching the outcome of a human decision does not guarantee an alignment with the cognitive processes that led to it.

To probe this question, we turn to the Odd-One-Out (O1O) task (Crutch et al., 2009; Sinapov & Stoytchev, 2010; Hebart et al., 2019; 2023), a cornerstone of cognitive psychology for evaluating conceptual representation and reasoning. In this task, a subject is presented with three objects and must identify the one that is least similar to the other two. This seemingly simple judgment reveals deep insights into the criteria, referred as cognitive dimensions, humans use to structure their conceptual world. Despite models have powerful capabilities, their accuracy in predicting human choices is surprisingly limited. This finding suggests that a fundamental gap exists between the models learned representations and the nuanced, context-dependent judgments characteristic of human cognition. The dominant fine-tuning approach, which relies on aligning models with raw behavioral data (i.e., what humans choose), appears insufficient to capture the richness of the human cognitive landscape (i.e., why they choose it).

Our key insight is that bridging this gap requires moving beyond mere behavioral mimicry to incorporate the foundational principles of human judgment. We hypothesize that fine-tuning an MLLM on data enriched with the core cognitive dimensions underlying human decisions will foster a more profound alignment. To this end, we propose a novel, data-driven methodology. We begin with the THINGS dataset (Hebart et al., 2019; 2023), a large-scale collection of O1O judgments. While the dataset provides the behavioral outcomes, the specific cognitive dimension for each trial is latent. By employing a jackknife (Mahner et al., 2025) procedure inspired by recent neuro-computational studies, we successfully infer the most probable cognitive dimension (e.g., man-made vs. natural, animal vs. non-animal) for each of triplets. Subsequently, we transform this triplet data with cognitive dimension into a rich, natural language format suitable for instruction tuning.

Using this newly crafted, cognitively-informed dataset, we fine-tune the Qwen2.5-VL-7B-Instruct model (Bai et al., 2025). Our experiments yield compelling results across multiple evaluation axes. First, our fine-tuned model outperforms the compared models in predicting human choices on the O1O task. Second, on a large-scale holdout set of over 90,000 trials sampling from 48 objects, our model demonstrates substantially higher consistency with human judgment patterns. Intriguingly, this enhanced human-like alignment does not come at the cost of general capabilities; the model's performance improves on the MMMU (test) (Yue et al., 2024a) and MMMU-Pro (Yue et al., 2024b) benchmarks and only marginally decreases on MMMU (val), indicating a favorable trade-off. Most notably, a searchlight representational similarity analysis (RSA) (Kriegeskorte et al., 2008) reveals that the fine-tuned model's internal representations show significantly increased alignment with neural activity in brain regions critical for planning and decision-making, such as the prefrontal cortex. This neuroscientific evidence provides a potential explanation for the observed behavioral improvements, suggesting our method encourages the model to develop representations that are not only behaviorally but also neurologically more aligned with humans.

Our contributions, therefore, present a promising new direction for developing MLLMs that are not just high-performing but are also more verifiably and fundamentally human-like.

## 2 RELATED WORK

**Multimodal Large Language Models.** The landscape of artificial intelligence has been reshaped by the extension of Large Language Models (LLMs) into the multimodal domain. MLLMs, such as LLaVA (Liu et al., 2023), MiniGPT-4 (Zhu et al., 2023), and the Qwen-VL series (Bai et al., 2023; Wang et al., 2024; Bai et al., 2025), have achieved unprecedented success by integrating powerful vision encoders with pre-trained LLMs. The dominant architecture typically involves a visual backbone (e.g., ViT (Dosovitskiy et al., 2020)) that processes images, a projection module that maps visual features into the language model's embedding space, and the LLM itself, which acts as the core reasoning engine. The training paradigm usually consists of two stages: an initial vision-language alignment pre-training on large-scale image-text pairs, followed by instruction fine-tuning on a curated set of multimodal conversational data to elicit desired behaviors. While this paradigm has proven effective for a wide range of tasks, our work diverges by focusing on a more cognitively-grounded fine-tuning objective, moving beyond standard instruction following.

**Odd-One-Out Task.** The O1O task has long been a staple in cognitive science for its effectiveness in revealing the structure of human conceptual knowledge (Du et al., 2025) without relying on verbal labels by analyzing patterns of choices, researchers can map out the psychological space of objects. The THINGS dataset (Hebart et al., 2019; 2023) represents a landmark effort in this area, providing a large-scale, high-quality benchmark of human O1O judgments. This dataset has been instrumental in evaluating the human-likeness of computational models of vision and semantics. Our work leverages this rich dataset not only as a benchmark but as a source from which to extract latent cognitive dimensions, turning a classic psychological experiment into a novel resource for fine-tuning the next generation of MLLMs.

**Aligning with Human Behavior and Cognition.** A growing body of research seeks to align more closely with human behavior and cognitive patterns in Figure 1. A prominent example is Reinforcement Learning from Human Feedback (RLHF) (Wainwright & Lowe, 2023), which fine-tunes models based on human preferences for generated outputs. Other studies have used behavioral data more directly, for instance, by training models to predict human choices in economic games or moral dilemmas. These approaches primarily focus on mimicking the outcomes of human decisions. Our

research builds upon this foundation but makes a crucial distinction: we argue that true human-like intelligence requires aligning with the underlying cognitive processes – the why behind a decision, not just the what. Instead of using raw behavioral traces, we enrich the training data with explicit representations of human cognitive dimensions, aiming for a deeper, more principled alignment.

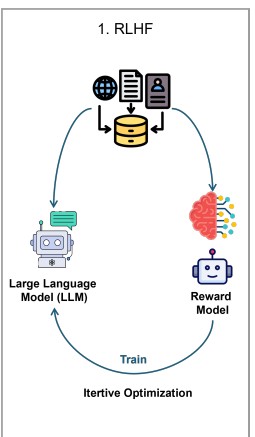 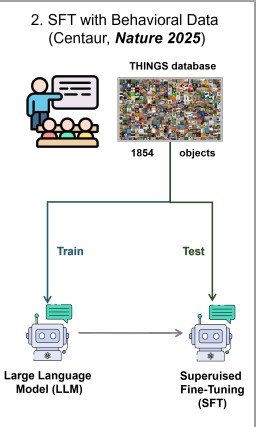 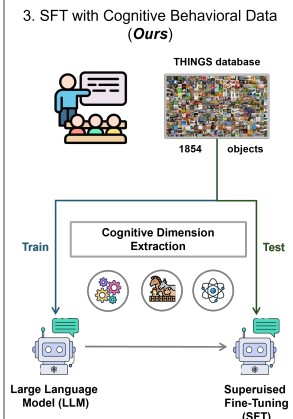

Figure 1: **Comparison of Model Alignment Methodologies.** This figure illustrates the evolution of alignment techniques aimed at making LLMs more human-like. (1) Reinforcement Learning from Human Feedback (RLHF) (Schulman et al., 2017; Rafailov et al., 2023; Liang, 2025) aligns models with human preferences by training a reward model on human-ranked outputs. (2) Supervised Fine-Tuning (SFT) with Behavioral Data aligns models directly on human behavioral examples, such as decisions in economic games or moral dilemmas. Both of these methods primarily focus on mimicking human outputs (Binz et al., 2025). (3) Our proposed method, different from previous work (Hebart et al., 2020; Sucholutsky et al., 2023; Zheng et al., 2018; Muttenthaler et al., 2023b;a), SFT with Cognitive Behavioral Data, represents a key distinction. Instead of merely using raw behavioral traces, we enrich the training data with explicit representations of the underlying human cognitive dimensions. This approach aims for a deeper, more principled alignment with human cognitive processes rather than just the final decisions.

# 3 METHODS

To integrate core cognitive dimension with corresponding triplets, we design a structured generation pipeline shown in Figure 2. This pipeline consists of two stages: first, identifying the underlying cognitive dimension driving human judgment, and second, incorporating this dimension into the LLM prompt to guide reasoning.

**Stage 1: Inferring Core Cognitive Dimensions.** The first stage focuses on extracting the specific latent cognitive dimensions (e.g., metallic or artificial, food-related) that humans utilize to make O1O judgments. We leverage the THINGS database (Hebart et al., 2019; 2023), a curated collection of 26,107 images representing 1,854 unique objects To extract these dimensions, we utilize Sparse Positive Object Similarity Embedding (SPoSE) (Zheng et al., 2018) combined with Jackknife re-sampling strategy (Mahner et al., 2025).

    i. Embedding Initialization: For each triplet, we obtain the representational embedding of 66 dimensions (Hebart et al., 2023). We first compute the baseline probability of the target object being the odd-one-out using the full embedding via the softmax function.

    ii. Jackknife Resampling: To identify the "core" dimension, we iteratively prune each of the 66 dimensions one at a time. For each iteration, we re-compute the probability of the target object using the remaining 65 dimensions.

    iii. Scoring: We calculate the variation (absolute difference) between the baseline probability and the pruned probability. This variation serves as the importance score for the pruned dimension.

iv. Selection: The dimension corresponding to the maximum variation score is identified as the core cognitive dimension, as its removal causes the most significant divergence from the original decision.

**Stage 2: Transforming Core Cognitive Dimensions.** Once we extract core cognitive dimensions, the second stage is to use them to guide the reasoning of a LLM. We achieve through a structured prompt approach shown in Figure 2. For each triplet, we take the explicit semantic-level cognitive dimension identified in Stage 1 and incorporate it directly into the LLM prompt. We provide the LLM with triplet brief descriptions. At the end of prompt, we add strong instruction with **"Think carefully and analyze in terms of core cognitive dimension"**. This instruction compels LLM to perform a deep, causal reasoning process based on the a priori human cognition, moving beyond superficial lexical similarities. The structured and enhanced prompt forces LLM to articulate why an object is the odd-one-out, leading to more aligned and interpretable outputs.

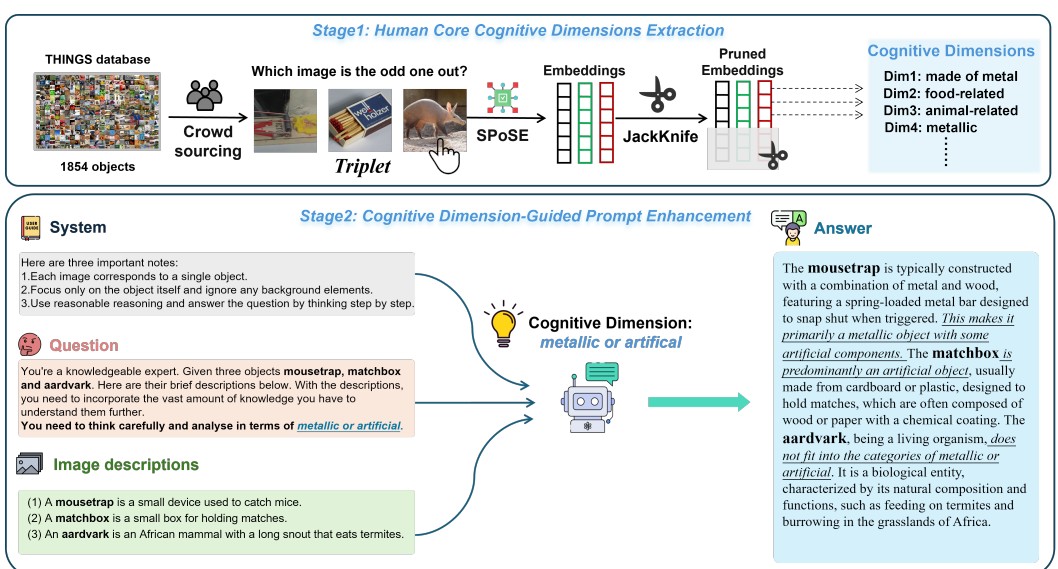

Figure 2: **Two-Stage Pipeline to Integrate Core Cognitive Dimension.** This figure illustrates our methodology for creating a cognitive-enhanced dataset for alignment. **Stage 1 shows the process of Human Core Cognitive Dimensions Extraction.** We use a crowdsourcing setup on image triplets from the THINGS database to collect human O1O judgments. We then use SPoSE and Jackknife analysis on their corresponding image embeddings to systematically extract the underlying core cognitive dimensions (e.g., metallic, food-related, animal-related). **Stage 2 demonstrates Cognitive Dimension-Guided Prompt Enhancement.** For a given triplet, we explicitly infuse a specific cognitive dimension into the model prompt. This directs the model to analyze the objects—in this example, a mousetrap, matchbox, and aardvark—along that precise dimension. This structured approach allows the model to generate a detailed, human-aligned, and explainable analysis content, effectively transforming a simple visual task into a cognitively-guided reasoning problem.

**Visual Instruction Tuning** We use Low-Rank Adaptation (Hu et al., 2022) approach with multi-task mixed training strategy for visual instruction tuning to achieve preserved performance. For architecture, we follow Qwen-2.5-VL-Instruct to adopt the most general framework, i.e., a vision encoder (Dosovitskiy et al., 2020), a projector, and a LLM (Bai et al., 2025). Low-rank Adaptation is a parameter-effective fine-tuning method that freezes the pretrained model weights and injects trainable rank decomposition matrices into each layer of the Transformer architecture (Vaswani et al., 2017), greatly reducing the number of trainable parameters for downstream tasks. We conduct preliminary experiments between full and LoRA fine-tuning. The results demonstrates that applying LoRA to the linear layers in projector and LLM achieves full fine-tuning level performance with less training times. Continue training (Zhou et al., 2024) on the sequential tasks data may cause model to converge to suboptimal local minima with poor performance due to distribution shift across tasks. We randomly shuffle the training data to maximize effectiveness of regularization data and adopt

placing all the image tokens in front of the prompt, while maintaining the placeholders within the text, denoted as "In-the-front" format.

**Human Consistency with MLLMs by Comparing Behaviors.** We evaluate human consistency from two aspects, one is O1O accuracy and the other is representational similarity matrix (RSM) correlation for the 48 objects by calculating the choice probability of each object pair.

For the O1O accuracy, we compare human true choices with model predictions in the held-out data as follows:

$$Acc_{O1O} = \frac{1}{n} \sum^n \begin{cases} 1, h_c = m_c \\ 0, h_c \neq m_c \end{cases} \tag{1}$$

where $n$ corresponds to the number of held-out data; $h_c$ to the human true choice and $m_c$ to the model prediction.

To measure the RSM correlation, following Rajalingham et al. (2018) work, we compute the Pearson Correlation on the behavioral RSMs from the human ($h$) and model ($m$) and then divide that raw Pearson Correlation by the geometric mean of the split-half internal reliability measured for each system as follows:

$$\tilde{\rho}(m, h) = \frac{\rho(RSM_m, RSM_h)}{\sqrt{\rho\left(RSM_m^{half_1}, RSM_m^{half_2}\right)\rho\left(RSM_h^{half_1}, RSM_h^{half_2}\right)}} \tag{2}$$

where $RSM_m^{half_1}$ and $RSM_m^{half_2}$ are computed by using the split-half behavioral data of triplets of the typical objects, and similar for human $RSM_h^{half_1}$ and $RSM_h^{half_2}$.

**Searchlight RSA.** For fMRI, local cerebral RSMs were computed in subject space within a grey-matter spherical region (6 mm diameter) centered at each voxel location. RSA (Kriegeskorte et al., 2008) assessed the Pearson correlation r between the local cerebral RSM and each kind of the model RSMs.

## 4 EXPERIMENTS

### 4.1 DATA

**Mixture Dataset.**

We use a mixture dataset composed primarily of the THINGS dataset (triplet O1O task), supplemented with additional datasets to preserve the generalization capability and prevent excessive specialization to specific domains. THINGS is a large behavioral dataset of 4.70 million unique triplet responses crowdsourced from 12,340 human participants for 1854 natural object images. Images used for collecting human responses in the triplet O1O task are taken from the THINGS object concept and image database, which is a collection of natural object images. We choose only the first 80,000 triplets with corresponding core cognitive dimension for each one. For additional datasets, we meticulously select single-image and multi-image scenarios (Alayrac et al., 2022; Jiang et al., 2024; Li et al., 2023) data from previous open-source datasets including COCO (Lin et al., 2014), ALLaVA-4V (Chen et al., 2024),

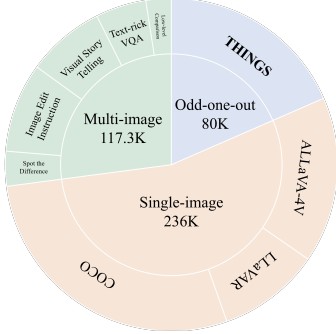

Figure 3: **Single-image**: COCO, ALLaVA-4V, LLaVAR. **Multi-image**: Spot the Difference, Image Edit Instruction, Visual Story Telling, Text-rich VQA, and Low-level Comparision. **O1O**: THINGS.

LLaVAR (Zhang et al., 2023b), and M4-Instruct (Li et al., 2024) as regularization data.

We exhibit a data overview of mixture dataset in Figure 3 and show case task examples in Figure 4. For the single-image data, they are consist of Caption and VQA tasks from rewritten COCO, Vision-FLAN (Xu et al., 2024), or LAION (Schuhmann et al., 2022) raw data. For the multi-image data, they are all from 5 tasks which are Spot the Difference (Jhamtani & Berg-Kirkpatrick, 2018; Johnson et al., 2017), Image Edit Instruction Zhang et al. (2023a); Bodur et al. (2024), Visual Story Telling (Ravi et al., 2021; Huang et al., 2016), Text-rich VQA (Mathew et al., 2020), and Low-level Comparision (Fu et al., 2023; Sundaram et al., 2024) in M4-Instruct. We provide detailed data statistics in Appendix A.2

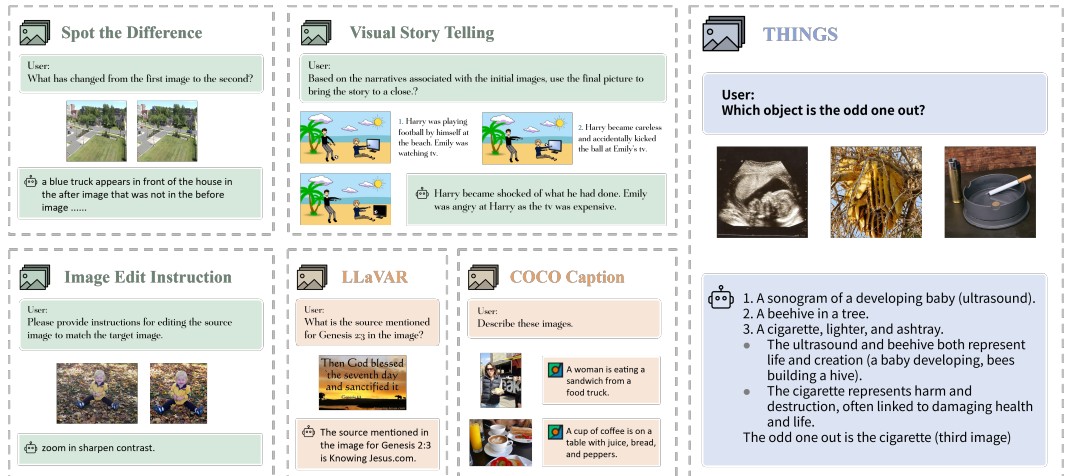

Figure 4: **Task examples of mixture dataset.**

**Brain Dataset.** We use Natural Scenes Dataset (NSD) (Allen et al., 2022) as the brain data, recognized as the largest neuroimaging dataset linking brain insights with artificial intelligence, involves richly sampled fMRI data from 8 subjects. Across 30-40 MRI sessions, each subject observed between 9,000-10,000 distinct natural scenes using whole-brain gradient-echo EPI at 1.8 mm isotropic resolution and 1.6 s TR during 7T scanning. Image stimuli are selected from the COCO dataset, with corresponding captions retrievable using COCO ID. To assess the correlation between humans neural responses and MLLMs representations stimulated in the same, the stimulations for each participant are chosen as the test set (because the searchlight RSA don't need to train). Additionally, fMRI responses linked to the stimulations across subjects S1, S2, S5, and S7 are earmarked for subsequent analysis (because subjects S3, S4, S6, and S8 did not complete the full fMRI data acquisition).

**Benchmark Dataset.** We evaluate our models from three aspects, generality in downstream tasks, consistency between humans and models by comparing behaviors, and searchlight RSA with visualizing on the brain. For the first generalization performance aspect, we choose two challenging benchmarks: MMMU, a benchmark designed to evaluate multimodal models on massive multi-discipline tasks demanding college-level subject knowledge and deliberate reasoning; and MMMU-Pro, a more challenging benchmark with more stringent assessment methodologies to evaluate multimodal models intrinsic understanding and reasoning capabilities. To assess behavior consistency, our preference is for the held-out THINGS validation set which is a resource designed to encompass 1,854 living and non-living objects based on their practical usage in daily life, and comprehensive triplets sampling on 48 objects. The final aspect for searchlight RSA to measure the correlation between humans neural responses and MLLMs representations, we use the brain dataset as mentioned above.

### 4.2 EXPERIMENTAL SETUP

Constrained by available computational resources, We use ∼7B series MLLMs to conduct experiments verifying our approach. We choose state of the art MLLMs, Qwen-2.5-VL-7B-Instruct, as baseline among open-source MLLMs in the ∼7B parameters range and fine-tune it with different parts of mixture dataset, resulting three distinct models for the following comparisions.

- **CogAligner (Baseline + Mixture dataset).** Fine-tune Qwen-2.5-VL-7B-Instruct with the entire mixture dataset.
- **BehavImitator (Baseline + Mixture dataset without core cognitive dimensions in THINGS).** Fine-tune Qwen-2.5-VL-7B-Instruct with mixture dataset without integrating core cognitive dimensions into THINGS part.
- **Qwen-2.5-VL-7B-Instruct-NT (Baseline + Mixture dataset without THINGS).** Fine-tune Qwen-2.5-VL-7B-Instruct with mixture dataset without THINGS.

We fine-tune each model for one epoch, i.e., a single pass over respective training data. All fine-tuning uses TRL (von Werra et al., 2020) with consistent hyperparameters for fair comparison. We also evaluate recent a foundation model of human cognition, Centaur (Binz et al., 2025) published on Nature, to compare with our CogAligner model.

To verify our method on other MLLMs, we extend experiments. We currently choose Gemma3-12B-it as baseline series and fine-tune them with mixture dataset. These finetuned models with mixture dataset are named as CogAligner$_{model}$.

### 4.3 MAIN RESULTS

**Behavioral Accuracy on O1O Task.** To evaluate human consistency with MLLMs, we compare behavioral accuracy of O1O on full validation set in the Figure 5. Few-shot prompting strategy is adopted in Centaur (n = 3, 5 shots) to infer the O1O by leveraging prior examples as context, as recommended in the Marcel Binz implementation. But zero-shot prompting strategy for others to generate directly. The extended comparative experiments results are in Table 1.

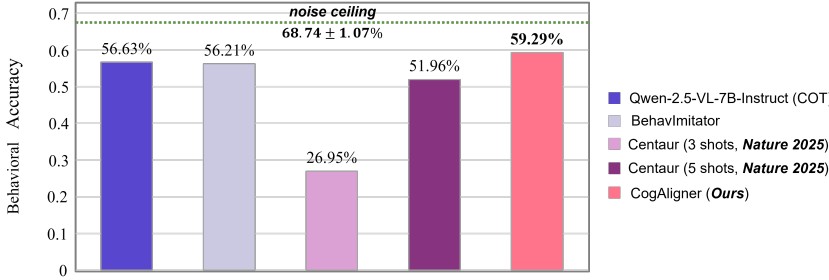

Figure 5: **Average accuracy are reported across total 453,642 triplets in the held-out THINGS validation set.** The instruction prompts in Centaur (n = 3, 5 shots) is constructed by providing 3 or 5 in-context examples. Qwen-2.5-VL-7B-Instruct (COT) adopts chain-of-thoughts (Wei et al., 2022) prompt, is constructed by appending "Let's think step by step" to the original prompt. The noise ceiling accuracy: 68.74 $\pm$ 1.07% (Hebart et al., 2023).

Table 1: **Extended Comparative Experiments.** Each row shows one model (COT) performance on O1O task reflected in choice accuracy. The CogAligner$_{model}$ indicates the O1O accuracy of aligned model initialized from base model in the same row. The noise ceiling is 68.74 $\pm$ 1.07%.

| Model | Baseline | CogAligner$_{model}$(ours) |
|---|---|---|
| Gemma3-12B-it | 55.766% | 57.279% |

**Human Consistency by Comparing RSMs.** Furthermore, we assess the behavioral consistency between humans and MLLMs by comparing RSMs with Pearson Correlation metric for the 48 objects. The results and RSMs are shown in Figure 6. We find the human consistency of CogAligner achieves 80.93% and outperforms others. We also observe interesting results from Centaur (3, 5 shots), demonstrating that different shots prompting influence the human consistency.

**Generalization Performance on Downstream Tasks.** We evaluate the generalization ability using the MMMU and MMMU-Pro dataset with lmms-eval. Table 2 summarizes the accuracy of Qwen-

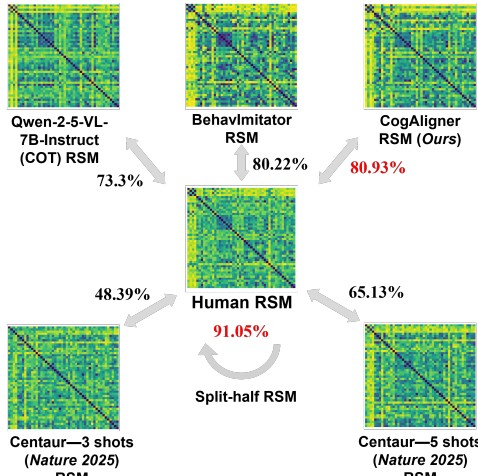

Figure 6: **Human Consistency between human and MLLMs.** We quantify human self-consistency by computing the Pearson Correlation between the first half and the second half of human RSM. And human self-consistency serves as the noise ceiling which is **91.05%**. For each human consistency, 95% confidence intervals are estimated using 1000 bootstrap resamples (Human: 95%CI [0.8994, 0.9204], Qwen-2.5-VL-7B-Instruct: 95%CI [0.6951, 0.768], BehavImitator: 95%CI [0.7712, 0.8304], CogAligner: 95%CI [0.7786, 0.8373], Centaur (3 shots): 95%CI [0.4314, 0.5339], Centaur (5 shots): 95%CI [0.6121, 0.688]).

2.5-VL-7B-Instruct, Qwen-2.5-VL-7B-Instruct-NT, BehavImitator, adn CogAligner. We also add extended experiments of on MMMU and MMMU-Pro dataset on other MLLMs in Table 2. We observe that CogAligner achieves the best performance on MMMU-Pro Vision, surpassing the baseline by 0.58%, while BehavImitator and Qwen-2.5-VL-7B-Instruct-NT also show improvements of 1.45% and 1.04% respectively. These results on MMMU and MMMU-Pro indicate that our method not only maintain generality in downstream tasks but can even achieve performance improvements without carefully adjusting model hyperparameters.

Table 2: **Generalization performance on MMMU and MMMU-Pro.** We report the average accuracy on MMMU (validation and test sets) and MMMU-Pro (standard and vision). Details in Appendix A.5

| Model | $MMMU_{val}$ | $MMMU_{test}$ | MMMU-Pro$_{standard}$ | MMMU-Pro$_{vision}$ |
|---|---|---|---|---|
| Qwen-2.5-VL-7B-Instruct | **51.78**% | 40.90% | 36.13% | 33.24% |
| Qwen-2.5-VL-7B-Instruct-NT | 49.33% | 45.40% | **37.63**% | 32.78% |
| BehavImitator | 50.0% | **46.40**% | 37.23% | 32.37% |
| CogAligner (ours) | 49.44% | 46.10% | 37.34% | **33.82**% |
| Gemma3-12B-it | **47.11**% | 35.60% | **31.33**% | 4.51% |
| CogAligner$_{Gemma3-12B-it}$ | 44.56% | **41.50**% | 30.87% | **14.80**% |

**Visualizing representative Odd-One-Out Task Example.** We present a representative example for O1O task among the validation set in Figure 7, identifying the most dissimilar object among triplet objects (contour, rocket, and maggot), where the core human cognitive dimension is transportation-related or movement-related. Our empirical findings reveal a significant discrepancy in the reasoning capabilities. Specifically, Qwen2.5-VL-7B-Instruct model not only fails to provide the correct answer but also adopts a reasoning perspective that is misaligned with the human cognitive dimension. Conversely, while the Centaur model arrives at the correct conclusion, it lacks the necessary explanatory capacity, raising questions about whether its success is due to genuine understanding or merely superficial mimicry of human output. In sharp contrast, our CogAligner model correctly identifies the most dissimilar object and, more importantly, provides a detailed and well-aligned explanation that directly leverages the human core cognitive dimension which is transportation-related or movement-related. This example shows that CogAligner exhibits a higher degree of alignment with core human cognitive dimensions, enabling it to better predict human behavior and offering a more interpretable pathway to its reasoning. This example also shows that CogAligner has emerged with a deeper, human-like understanding.

**Visualizing Searchlight RSA.** We perform searchlight RSA between subjects and MLLMs, including Qwen-2.5-VL-7B-Instruct, Qwen-2.5-VL-7B-Instruct-NT, BehavImitator, and CogAligner,

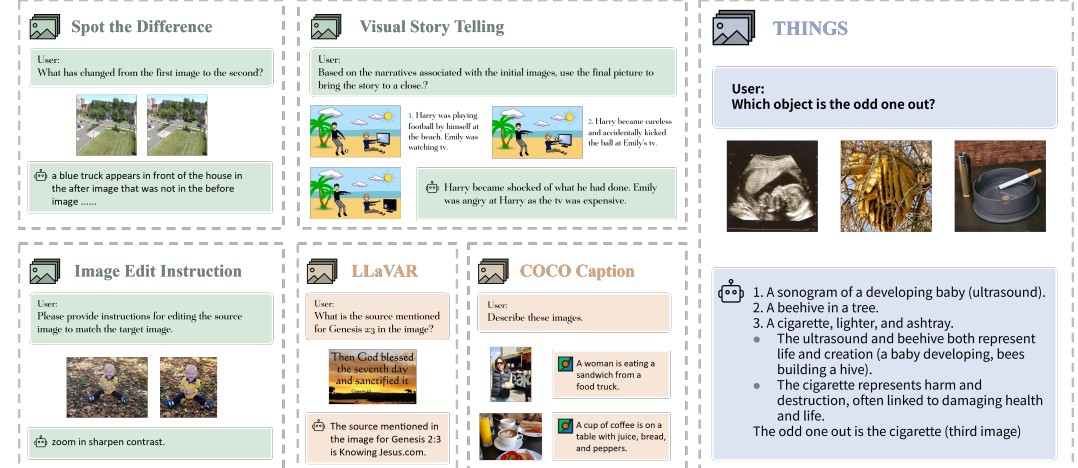

Figure 7: **A representative example study demonstrating the superior cognitive alignment of the CogAligner model.** The example is to identify the most dissimilar object among three items (wheel rim, rocket, maggot) and the core human cognitive dimension is **transportation-related or movement-related**. Qwen2.5-VL-7B-Instruct model provides an incorrect answer (B) and its reasoning is misaligned with the human core cognitive dimension, instead relying on a **stationary vs. dynamic** distinction. Centaur model, while providing the correct answer (C), offers no explanation, leaving its reasoning process opaque. Our CogAligner model not only gives the correct answer (C) but also provides a detailed reasoning process that is fully aligned with the core human cognitive dimension.

separately. To enhance the contrast of between the brain alignment of CogAligner and others, we visualize voxel-wise difference on Qwen-2.5-VL-7B-Instruct and CogAligner in Figure 8 via projection to the cerebral cortex (Gao et al., 2015). To each subject contrast result, we perform the two-sample Kolmogorov-Smirnov test. We observe increasing activation in brain regions associated with problem planning and decision-making, such as the prefrontal cortex, in the CogAligner.

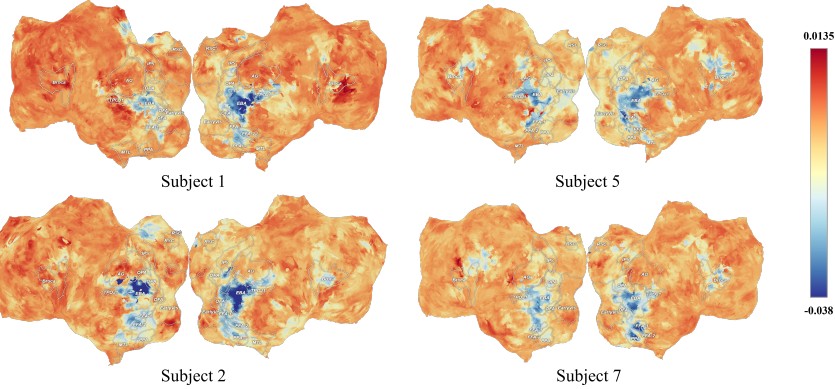

Figure 8: **Subject contrast results between Qwen-2.5-VL-7B-Instruct and CogAligner in searchlight RSA experiment.** Red indicates increased alignment of CogAligner relative to the Qwen-2.5-VL-7B-Instruct, while blue indicates decreased alignment on the bar. Two-sample Kolmogorov-Smirnov test to each subject contrast result (subject 1: $KS = 0.0095; p < 0.05$, subject 2: $KS = 0.0151; p < 0.05$, subject 5: $KS = 0.0151; p < 0.05$, subject 7: $KS = 0.0149; p < 0.05$)

## 5 CONCLUSIONS

In this work, we address the critical yet underexplored challenge of bridging the gap between human-level performance and human-like cognitive processes in MLLMs. We argue that conventional alignment techniques, which primarily focus on mimicking behavioral outcomes, are insufficient to close this gap. Our key contribution is a novel fine-tuning paradigm, SFT with Cognitive Behavioral Data, which moves beyond behavioral mimicry to align models with the underlying cognitive dimensions that guide human judgment. Our methodology successfully extracts latent cognitive principles from large-scale human behavioral data and explicitly integrates them into the model's instruction-tuning process.

The resulting model, CogAligner, demonstrates markedly superior alignment with human cognition. Empirically, CogAligner not only achieves higher accuracy in predicting human choices but also exhibits internal representations that are significantly more consistent with human judgmental patterns, as measured by RSA. Crucially, this enhanced cognitive alignment maintains and improve performance on challenging downstream multimodal benchmarks, such as MMMU and MMMU-Pro, without sacrificing generality.

Furthermore, our searchlight RSA analysis provides neuroscientific evidence corroborating these findings, revealing that CogAligner's representations are more aligned with neural activity in key decision-making regions of the human brain, including the prefrontal cortex. This suggests our approach encourages the development of a more principled and neurologically plausible reasoning framework. Our findings chart a promising new course for developing MLLMs that are not only performant but also more fundamentally and verifiably aligned with the nuances of human intelligence.

## 6 LIMITATIONS AND FUTURE DIRECTIONS

Our current experimental verification focuses primarily on the Qwen-2.5-VL-7B-Instruct model. While we extend our verification to Gemma3-12B-it model to demonstrate the generalizability of our approach across different model families, we have not yet verified the efficacy of our method on large-scale models (e.g., 32B, 70B) or closed-source models (e.g., GPT-5, Gemini 3). It remains an open question whether the alignment with human cognitive dimensions emerges naturally with scale or if the benefits of our method scale proportionally with model scale.

The core of our method relies on THINGS database, which contains judgments on 1,854 unique natural objects. Consequently, the extracted 66 cognitive dimensions may not fully encompass the complexity of human cognition required for abstract reasoning or understanding complex temporal events in videos. The current framework is therefore limited to visual object understanding and may require adaptation for broader multimodal tasks.

Our two-stage pipeline relies on the availability of high-quality human behavioral data (e.g., Odd-One-Out triplets) to infer latent cognitive dimensions via SPOSE and Jackknife. This dependence limits the immediate scalability of the method to domains where such rich behavioral datasets are unavailable.

## ETHICS STATEMENT

We confirm that all authors have read and comply with the ICLR code of ethics `https://iclr.cc/public/CodeOfEthics`.

## 7 REPRODUCIBILITY STATEMENT

We provide training details in Appendix A.3 and evaluation details in Appendix A.4.

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

## A  APPENDIX

### A.1  USE OF LLMS

The authors explicitly declare the use of a large language model (LLM) in the preparation of this manuscript. The LLM is utilized exclusively as an editing and proofreading tool to improve the grammar, syntax, and overall readability of the text.

Specifically, the LLM is employed for the following purposes:

- **Language Polishing:** Correction of grammatical errors, spelling mistakes, and punctua-tion.

- **Sentence Structure:** Refinement of sentence and paragraph structure to enhance clarity and conciseness.

- **Readability:** Suggestions for more formal and academic phrasing to align with scholarly writing standards.

The LLM was not used for any content-generating tasks, including but not limited to: ideation, methodology development, result analysis, or the generation of core scientific arguments. All con-tributions, experimental design, and analytical insights are solely the work of the authors. The authors bear full responsibility for the content and integrity of this paper.

## A.2 DATA STATISTICS

Table 3: **Detailed Mixture Dataset Statistics.**

| Task | Dataset | Training Samples | Validation Samples |
|---|---|---|---|
| **Single-image Scenarios** | | | |
| Caption (13.5K) | COCO | 12,150 | 1,351 |
| ALLaVA-4V (69.98K) | ALLaVA-VFLAN | 17,991 | 1,999 |
| | ALLaVA-LAION | 44,991 | 4,999 |
| LLaVAR (43.167K) | LLaVAR-GPT4 | 38,850 | 4,317 |
| **Multi-image Scenarios** | | | |
| Spot the Difference (14.659K) | Spot-the-Diff | 9,696 | 1,078 |
| | CLEVR-Change | 3,690 | 195 |
| Image Edit Instruction (23.013K) | MagicBrush | 17,601 | 1,956 |
| | IEdit | 3,283 | 173 |
| Visual Story Telling (32.941K) | AESOP | 6,569 | 346 |
| | IEdit | 23,423 | 2,603 |
| Text-rich VQA (21.387K) | WebQA | 8,871 | 467 |
| | TQA | 7,836 | 413 |
| | OCR-VQA | 1,805 | 95 |
| | DocVQA | 1,805 | 95 |
| Low-level Comparison (10.682K) | Dreamsim | 9,613 | 1,069 |
| **Behavioral Tasks** | | | |
| O1O (80K) | THINGS | 64,000 | 16,000 |

## A.3 TRAINING DETAILS

In this section, we present all the hyperparameters we use to fine-tune in Table 4. These hyperparameter settings are shared across all finetuned models mentioned in this paper. All the training processes are conducted using Transformers, PEFT, and TRL libraries.

Table 4: **Hyperparameter Settings for fine-tuning.**

| Hyperparameter | Value |
|---|---|
| seed | 42 |
| LoRA Rank | 8 |
| LoRA $\alpha$ | 32 |
| LoRA dropout | 0.05 |
| LoRA bias | No |
| learning rate | 0.00005 |
| epoch | 1 |
| dtype | bfloat16 |
| attn implementation | sdpa |
| device numbers | 6 |
| gradient accumulation | 8 |
| train batch size (per device) | 4 |
| train batch size (total) | 192 |
| eval batch size (per device) | 4 |
| eval batch size (total) | 192 |
| padding side | right |
| max pixels | 451,584 |
| min pixels | 12,544 |

### A.4 EVALUATION DETAILS

In this section, we introduce benchmarks of generalization of performance, O1O accuracy, human consistency, and searchlight RSA in details.

**Generalization of Performance.** We use lmms-eval open-source evaluation suite of large multi-modal models to test performance of MLLMs on MMMU (Val and test) and MMMU-Pro (standard and vision). For MMMU and MMMU-Pro, we use zero shot and bfloat16 model dtype and default settings(e.g., temperature=0.01, flash-attention-2) in lmms-eval.

**O1O Accuracy and Human Consistency.** we use **model.generate** with Transformers library (Wolf et al., 2020) to sample outputs of MLLMs. The most important hyperparameters for inferring on THINGS validation set are **temperature**, **top p**, and **top k**. To achieve stable outputs, we set the temperature to 0.01, the top p to 0.001, and the top k to 1. For other hyperparameters, We use bfloat16 (dtype) and sdpa/flash attention (attn implementation) with seed 42.

**Searchlight RSA.** We use our implementation with Pytorch to accelerate compute searchlight RSA. We encourage the use of our released codes for searchlight RSA to save times.

### A.5 MMMU AND MMMU-PRO DETAILED EVALUATION RESULTS

Table 5: **Detailed Generalization Performance on MMMU Val.**

| Model | Art and Design | Business | Health and Medicine | Humanities and Social Science | Science | Tech and Engineering | Average |
|---|---|---|---|---|---|---|---|
| Qwen-2.5-VL-7B-Instruct | **69.167%** | 41.333% | 55.333% | **73.333%** | 41.333% | 41.905% | **51.778%** |
| Qwen-2.5-VL-7B-Instruct-NT | 66.667% | **42.667%** | 52.0% | 66.667% | 36.0% | 41.905% | 49.333% |
| BehavImitator | **69.167%** | 40.667% | **56.667%** | 69.167% | 36.667% | 39.524% | 50.0% |
| CogAligner (ours) | 66.667% | 39.333% | 53.333% | 64.167% | 40.667% | **41.905%** | 49.444% |
| Gemini-12B-it | **65.0%** | 37.333% | **48.667%** | **70.833%** | 35.333% | **37.619%** | **47.111%** |
| CogAligner$_{\text{Gemma3-12B-it}}$ | 55.0% | **38.667%** | 43.333% | 65.833% | **41.333%** | 33.81% | 44.556% |

Table 6: **Detailed Generalization Performance on MMMU Test.**

| Model | Art and Design | Business | Health and Medicine | Humanities and Social Science | Science | Tech and Engineering | Average |
|---|---|---|---|---|---|---|---|
| Qwen-2.5-VL-7B-Instruct | 55.4% | 45.2% | 46.1% | 56.2% | 34.1% | 30.2% | 40.9% |
| Qwen-2.5-VL-7B-Instruct-NT | 59.5% | **47.9%** | 50.9% | 62.6% | 39.4% | 34.2% | 45.4% |
| BehavImitator | **60.5%** | 46.7% | **52.6%** | 64.3% | **40.8%** | 35.4% | **46.4%** |
| CogAligner (ours) | 58.6% | 45.1% | 51.9% | 65.2% | 39.8% | **36.6%** | 46.1% |
| Gemini-12B-it | 39.5% | **45.7%** | 35.2% | 39.9% | 32.6% | 30.0% | 35.6% |
| CogAligner$_{\text{Gemma3-12B-it}}$ | **55.3%** | 38.2% | **43.4%** | **62.5%** | **33.3%** | **36.1%** | **41.5%** |

Table 7: **Detailed Generalization Performance on MMMU-Pro Standard.**

| Model | Art and Design | Business | Health and Medicine | Humanities and Social Science | Science | Tech and Engineering | Average |
|---|---|---|---|---|---|---|---|
| Qwen-2.5-VL-7B-Instruct | 56.140% | 28.42% | 30.42% | **47.748%** | 31.615% | 31.415% | 36.127% |
| Qwen-2.5-VL-7B-Instruct-NT | 54.386% | **29.371%** | **34.615%** | 46.847% | **34.708%** | **33.333%** | **37.630%** |
| BehavImitator | **56.579%** | 28.322% | 34.266% | 44.595% | 34.021% | 33.094% | 37.225% |
| CogAligner (ours) | 55.263% | 28.322% | 34.266% | 46.847% | 34.021% | 33.094% | 37.341% |
| Gemini-12B-it | **50.877%** | **24.126%** | **26.573%** | 47.748% | 26.46% | 23.501% | **31.329%** |
| CogAligner$_{\text{Gemma3-12B-it}}$ | 47.807% | 20.28% | 25.524% | **51.351%** | **27.835%** | **23.741%** | 30.867% |

Table 8: **Detailed Generalization Performance on MMMU-Pro Vision.**

| Model | Art and Design | Business | Health and Medicine | Humanities and Social Science | Science | Tech and Engineering | Average |
|---|---|---|---|---|---|---|---|
| Qwen-2.5-VL-7B-Instruct | **46.491%** | 24.825% | 28.322% | **45.495%** | 31.959% | 29.496% | 33.237% |
| Qwen-2.5-VL-7B-Instruct-NT | 44.737% | 24.476% | 28.671% | 42.342% | 32.990% | 29.496% | 32.775% |
| BehavImitator | 45.175% | **26.923%** | 28.322% | 41.892% | 30.241% | 28.297% | 32.37% |
| CogAligner (ours) | 45.175% | 25.874% | **29.371%** | 43.243% | **34.021%** | **30.935%** | **33.815%** |
| Gemini-12B-it | 7.456% | 1.748% | 6.643% | 8.559% | 4.811% | 0.959% | 4.51% |
| CogAligner$_{\text{Gemma3-12B-it}}$ | **16.667%** | **14.685%** | **9.091%** | **21.622%** | **14.777%** | **14.149%** | **14.798%** |

## A.6  66 COGNITIVE DIMENSIONS

Table 9: **66 Dimensions Name.**

| axis | Dimension | axis | Dimension |
|---|---|---|---|
| 0 | metallic or artificial | 1 | food-related |
| 2 | animal-related | 3 | textile |
| 4 | plant-related | 5 | house-related or furnishing-related |
| 6 | valuable or precious | 7 | transportation-related or movement-related |
| 8 | body-related or people-related | 9 | wood-related or brown |
| 10 | electronics or technology | 11 | colorful or playful |
| 12 | outdoors | 13 | circular or round |
| 14 | paper-related or flat | 15 | sports-related or playing-related |
| 16 | tools or elongated | 17 | fluid-related or drink-related |
| 18 | water-related | 19 | oriented or many things |
| 20 | decay-related or grainy | 21 | white |
| 22 | coarse pattern or many things | 23 | red |
| 24 | long or thin | 25 | weapon-related or danger-related |
| 26 | black | 27 | household |
| 28 | feminine (stereotypical) | 29 | body part-related |
| 30 | tubular | 31 | music-related, hearing-related, or hobby-related |
| 32 | grid-related or grating-related | 33 | repetitive or spiky |
| 34 | construction-related or craftsman-related | 35 | spherical or voluminous |
| 36 | string-related or stringy | 37 | seating-related, standing-related, or lying-related |
| 38 | flying-related or sky-related | 39 | disgusting or slimy |
| 40 | elliptical or curved | 41 | sand-colored |
| 42 | green | 43 | bathroom-related or wetness-related |
| 44 | yellow | 45 | heat-related or light-related |
| 46 | beams-related or mesh-related | 47 | foot-related or walking-related |
| 48 | box-related or container | 49 | stick-shaped or cylindrical |
| 50 | head-related | 51 | upright, elongated, or volumous |
| 52 | pointed or spiky | 53 | child-related or cute |
| 54 | farm-related or historical | 55 | seeing-related, small, or round |
| 56 | medicine-related | 57 | dessert-related |
| 58 | orange | 59 | thin or flat |
| 60 | cylindrical, conical, or cushioning | 61 | coldness-related or winter-related |
| 62 | measurement-related or numbers-related | 63 | fluffy or soft |
| 64 | masculine (stereotypical) | 65 | fine-grained pattern |

