# OpenReview forum: "Beyond Behavioral Alignment: Leveraging Core Cognitive Dimensions for Enhanced Human-like MLLMs"
_ICLR.cc/2026/Conference — ICLR 2026 Conference Withdrawn Submission_

### Official Review · Reviewer_tnH7 · 2025-10-21

**Soundness:** 2
**Presentation:** 1
**Contribution:** 2
**Rating:** 2
**Confidence:** 3

**Summary:**

This paper proposes a new method to make MLLMs more human-like by aligning them with core cognitive dimensions rather than just imitating human behavior.
Using the Odd-One-Out (O1O) task and the THINGS dataset, the authors infer latent cognitive bases (like “natural vs. artificial”) from human judgments and integrate these dimensions into the model’s training process.
Their fine-tuned model, CogAligner, achieves higher consistency with human decisions, better alignment with neural activity in decision-related brain regions, and improved generalization on reasoning benchmarks.
The work demonstrates that incorporating cognitive principles enables models to move beyond behavioral mimicry toward neurologically and cognitively grounded human-like intelligence.

**Strengths:**

Interesting idea. I do believe that machines cannot only learn “what”, they need to also learn “why” and “how.” Current good performance of many large models is just a result of flooding the models with plenty of training data. VLMs still perform poorly on some fundamental visual tasks, which are quite easy for humans. I think models should build on a bottom-up style, mastering the basics before the downstream tasks.

**Weaknesses:**

- Only verify the idea on one model, Qwen-2.5-VL-7B. The performance improvement on this single model is not statistically significant. Additionally, how much do you think the approach can generalize to larger models and other model families?
- Not perfect in MMMU-Pro. Moreover, Qwen2.5-VL-7B reports its MMMU-Pro accuracy being 38.3 (https://mmmu-benchmark.github.io/#leaderboard) or 41.0 (https://huggingface.co/Qwen/Qwen2.5-VL-7B-Instruct). This paper reports 32.9, which is obviously lower. Could you explain the possible reasons?
- The writing of the current paper should be largely improved.
- The core part of the proposed method is to use LLMs to generate a gold answer using the key dimension recognized by Jackknife. Do you go through any human evaluation the generation quality? What is the accuracy of model on THINGS dataset with and without the key dimension as a hint?

**Questions:**

1. What are models’ brain regions (the second last sentence in the abstract)?
2. How do you generate your ground truth for training (i.e., the “Answer” in Fig. 2)? Is the “Answer” in Fig. 2 the same thing as the assistant message in Fig. 4?
3. What is RSM? The term is not defined in the whole paper. I assume it refers to representational similarity matrix, but you need to explicitly write it down. What is RDM in your Fig. 6?
4. How did you decide the proportion of dataset mixture in Fig. 3? Do you complete any ablation study on the influence of different proportions?
5. Could you please append the 66 dimensions in SPoSE in the manuscript?
6. Why the Centaur uses a few-shot setting while other models use a zero-shot setting? Is this a fair comparison?
7. Can you provide the STD or 95% confidence interval for the results in Fig. 6?

Minor suggestions and typos:
1. In the abstract, the meaning of RSA is not clear. Since it appears only once in the abstract, I suggest changing it directly to representational similarity analysis.
2. The organization of the method section should be improved. The writing in the method section is not clear. You mention Jackknife in the first paragraph (Line 143-150) and then mention it again in the second paragraph (Line 161-164). A better practice may be Stage 1, Stage 2, …
3. A typo: Line 241-242: “RSA analyses assessed” -> “RSA assessed”
4. A typo: Line 259: “THINGS object oncept” - > “THINGS object concept”
5. An additional “~” at line 323 before “7B”
6. Missing spaces: Line 367 “Centaur(3, 5 shots)”; Line 422 “MMMU-Pro.Table”

---

> ### Author Response · Authors · 2025-11-21
>
> We sincerely thank the reviewer for recognizing the novelty of our work and for the insightful feedback regarding the experimental scope and presentation. We have thoroughly revised the manuscript to address these concerns (changes highlighted in orange and red corrections for issues common to all reviewers.). Below, we provide detailed responses to each point.
>
> ## 1. Only verify the idea on one model, Qwen-2.5-VL-7B. The performance improvement on this single model is not statistically significant. Additionally, how much do you think the approach can generalize to larger models and other model families?
> ### Response:
> We appreciate this suggestion. While our initial submission focused on Qwen-2.5-VL-7B due to computational constraints, we agree that verifying generalization of the approach is critical.
>
> Our core method—leveraging core cognitive dimensions—is a data-driven approach independent of model architecture. The improvements stem from the quality of the cognitive alignment data rather than specific architectural tweaks.
>
> During the rebuttal period, we applied our method to Gemma3-12B-it. As shown in Table 1a and Table 1b below, the CogAligner approach consistently improves performance on the Odd-One-Out (O1O) task and maintains or improves generalization on MMMU and MMMU-Pro benchmarks compared to corresponding baseline. This confirms that our approach generalizes effectively to different model families and larger parameter scales.
>
> Table 1a. General Performance on MMMU and MMMU-Pro
> | Model | MMMU$_{val}$ | MMMU$_{test}$ | MMMU-Pro$_{standard}$ | MMMU-Pro$_{vision}$ |
> |----------------|-------|--------|----------|-------|
> | Gemma3-12B-it | 47.111% | 35.6% | 31.329% | 4.509% |
> | CogAligner$_{Gemma3-12B-it}$ | 44.556% | **41.5%** | 30.867% | **14.798%** |
>
> Table 1b. Odd-One-Out Accuracy
> | Model | Accuracy |
> |---------|-------|
> | Gemma3-12B-it | 55.766% |
> | CogAligner$_{Gemma3-12B-it}$ | **57.279%** |
>
> We have added these additional experiments verifying our approach on Gemma3-12B-it in Table 1&2 (highlighted red). We will do more verification experiments and will release results on openreview after these experiments finish.
>
> ## 2. Not perfect in MMMU-Pro. Moreover, Qwen2.5-VL-7B reports its MMMU-Pro accuracy being 38.3 (https://mmmu-benchmark.github.io/#leaderboard) or 41.0 (https://huggingface.co/Qwen/Qwen2.5-VL-7B-Instruct). This paper reports 32.9, which is obviously lower. Could you explain the possible reasons?
> ### Response:
> We utilized the **lm-evaluation-harness** to evaluate under default settings. We achieved a score of 32.9% for Qwen2.5-VL-7B-Instruct on MMMU-Pro.
>
> The higher scores (38.3/41.0) reported on leaderboards are likely to utilize internal, highly optimized proprietary prompts, different system instructions or settings that are not fully public. We attempted to reproduce those numbers using **lmms-eval** but obtained results lower than 38.3/41.0. Our retest results using **lm-eval** under default settings (e.g., temperature=0.01) on MMMU-Pro and MMMU are closer to leaderboard and have replaced raw in Table 2 (highlighted in red, revised manuscript).
>
> Our primary objective is not to enhance general model capabilities, but rather to align the MLLMs with core human cognitive dimensions via fine-tuning to achieve human-like behavior. Consequently, we utilize Odd-One-Out behavioral data instead of traditional datasets typically used for boosting general performance.
>
> However, we are surprised to observe that a model more closely aligns with human behavior does not suffer from alignment tax or performance degradation. On the contrary, it not only preserves its original capabilities but also achieves slight improvements on general benchmarks.

---

> ### Author Response · Authors · 2025-11-21
>
> ## 3. The core part of the proposed method is to use LLMs to generate a gold answer using the key dimension recognized by Jackknife. Do you go through any human evaluation the generation quality? What is the accuracy of model on THINGS dataset with and without the key dimension as a hint?
> ### Response:
> Before answering issue, we clarify our method process:
>
> Source of Dimensions: We do not "hallucinate" dimensions. We utilize the 66 cognitive validated dimensions from Martin Hebart [1,2].
> Core Cognitive Dimension: We use Jackknife method to identify the ground truth cognitive dimension among 66 dimensions for each triplet. The LLM is then used only to format this identified dimension into a natural language Chain-of-Thought (CoT) explanation (as shown in Fig. 2).
>
> Ablation Study (With/Without Dimension): We have already performed the requested ablation. The model "BehavImitator" in our paper represents the setting without the key cognitive dimension hint (training only on the triplet descriptions and correct choice).
>
> (1) We have gone through human evaluation the generation quality before LLM stabilizes and outputs analysis content (Fig.2 Stage 2 introduce).
>
> (2) The accuracy of model on THINGS dataset **with key dimension** as hint (**CogAligner**) is **59.29%**. However, the accuracy of model on THINGS dataset **without key dimension** as hint (**BehavImitator**) is 56.21%
>
> This comparison (shown in Fig. 5) explicitly demonstrates that the inclusion of the core cognitive dimension is the driver of performance improvement.
>
> ## 4. What are models' brain regions (the second last sentence in the abstract)?
> ### Response:
> We apologize for the ambiguity. This phrase is a metaphorical shorthand for "model representations that align with specific human brain regions."
>
> Specifically, via Searchlight Representational Similarity Analysis (RSA), we project the model's internal representations onto the human cortex. We found that CogAligner shows increased representational similarity to human neural activity in the prefrontal cortex (PFC)—a region associated with planning and decision-making—compared to the baseline.
>
> ## 5. How do you generate your ground truth for training (i.e., the “Answer” in Fig. 2)? Is the “Answer” in Fig. 2 the same thing as the assistant message in Fig. 4?
> ### Response:
> There is a distinction between the data generation pipeline(Fig.2) and the final training data(Fig.4).
>
> **Fig.2**: This illustrates how we construct the dataset. The "Answer" here refers to the Analysis Content generated by combining the extracted core cognitive dimension (from SPoSE/Jackknife) with image descriptions.
> **Fig.4**: This shows the final format used for Supervised Fine-Tune. The "Assistant Message" here includes the Analysis Content generated in Fig. 2.
>
> ## 6. What is RSM? The term is not defined in the whole paper. I assume it refers to representational similarity matrix, but you need to explicitly write it down. What is RDM in your Fig. 6?
> ### Response:
> RSM stands for Representational Similarity Matrix, and RDM stands for Representational Dissimilarity Matrix. We apologize for the inconsistency in Fig.6 labels. We have standardized all terminology to RSM (Representational Similarity Matrix) throughout the revised paper (highlighted in orange).
>
> ## 7. How did you decide the proportion of dataset mixture in Fig.3? Do you complete any ablation study on the influence of different proportions?
> ### Response:
> We utilized the first 80k samples from THINGS (split 64k train / 16k val). To prevent catastrophic forgetting and overfitting to the Odd-One-Out task, we mixed in general VQA and other tasks datasets at a ratio of approximately **4.4:1** (General:Odd-One-Out).
>
> While we do not perform a fine-grained ablation on the exact ratio due to computational limits, the strong performance on both the specific task (Odd-One-Out) and general benchmarks (MMMU, MMMU-Pro) suggests the chosen ratio is effective and robust.
>
> ## 8. Could you please append the 66 dimensions in SPoSE in the manuscript?
> ### Response:
> We agree this is important for reproducibility. We have added the full list of the 66 dimensions to Appendix A.6 Table 9 in the revised manuscript (highlighted in red). If our paper is accepted, we will also release source code on github and relevant data (e.g., 66 dimensions, training data).

---

> ### Author Response · Authors · 2025-11-21
>
> ## 9. Why the Centaur uses a few-shot setting while other models use a zero-shot setting? Is this a fair comparison?
> ### Response:
> We believe this comparison is fair and, in fact, favors the baseline (Centaur).
>
> The Centaur model (based on LLaMA-3.1-70B pretrained model) requires a few-shot setting to output valid formats because of our zero-shot tests with Centaur yielded incoherent outputs. Our model (CogAligner) operates in a zero-shot setting, which is generally considered more challenging than few-shot. By allowing Centaur to use few-shot examples to guide its reasoning, we are providing it with an advantage. Despite this advantage, CogAligner demonstrates superior alignment and interpretability, strengthening our claims.
>
> ## 10. Can you provide the STD or 95% confidence interval for the results in Fig. 6?
> ### Response:
> We have calculated the 95% confidence intervals for results of Fig.6. The 95% confidence intervals have been added in caption of Fig.6 (higlighted in revised manuscript) and shown in Table 2 below.
>
> Table 2. 95% Confidence Intervals for Human Consistency of Fig.6
> | Model | Human Consistency | Lower Bound | Upper Bound |
> |----------------|-------|--------|----------|
> | Human | 91.05% | 89.94% | 92.04% |
> | Centaur$_{3 shot}$ | 48.39% | 43.14% | 53.39% |
> | Centaur$_{5 shot}$ | 65.13% | 61.21% | 68.80% |
> | Qwen-2.5-VL-7B-Instruct(COT) | 73.30% | 69.51% | 76.80% |
> | BehavImitator | 80.22% | 77.12% | 83.04% |
> | CogAligner(ours) | 80.93% | 77.86% | 83.73% |
>
>
> ## 11. The meaning of RSA and the writing in the method section is not clear. “RSA analyses assessed, THINGS object oncept, ~, Centaur(3,5 shots), MMMU-Pro.Table” typos.
> ### Response:
> We thank the reviewer for the careful reading. We have corrected all identified typos (RSA definition, Jackknife placement, "object concept," etc.) and improved the organization of the Methods section to clearly delineate "Stage 1" and "Stage 2" as suggested. These revised results are highlighted in red.
>
> ### References:
> [1] Hebart, M. N., Zheng, C. Y., Pereira, F. & Baker, C. I. Revealing the multidimensional mental representations of natural objects underlying human similarity judgements. Nat. human behaviour 4, 1173–1185 (2020).
>
> [2] Hebart, M. N. et al. THINGS-data, a multimodal collection of large-scale datasets for investigating object representations in human brain and behavior. Elife 12, e82580 (2023).

---

> > ### Comment · Reviewer_tnH7 · 2025-11-24
> >
> > Thanks for the details. Most of the concerns have been addressed. There are some remaining:
> >
> > 4. I have a more high-level question: whether it is useful or even possible to map a concept from human cognition, such as prefrontal cortex, onto AI like LLMs. Since human information processing is very different from that of LLMs, wouldn’t each definition lack critical traits?
> > 5. Combining your response of (3), in my understanding, LLMs are used only to normalize the dimensions identified by Jackknife into certain fotmats (the answer in Fig. 2 and the assistant messages in Fig. 4), is this correct?
> > 6. Remember to explicitly write down "representational similarity matrix" before you first use "RSM". Currently I still did not see it in the paper.

---

> ### Author Response · Authors · 2025-11-24
> **Thank you for your suggestions!**
>
> We sincerely appreciate the time you spent reviewing our responses and the revised manuscript, as well as your valuable feedback.
>
> ## 4. I have a more high-level question: whether it is useful or even possible to map a concept from human cognition, such as prefrontal cortex, onto AI like LLMs. Since human information processing is very different from that of LLMs, wouldn’t each definition lack critical traits?
> ### Response:
> We thank the reviewer for this **thought-provoking** question. We fully agree with your perspective that AI, such as LLMs or MLLMs, differs fundamentally from the human information processing.
>
> However, despite these differences, we argue that pursuing **human alignment** remains scientifically significant.
>
> Prior work [1,2,3] indicates that, although the underlying mechanisms differ, advanced AI and human cognition can converge functionally. Focusing on aligning these functional aspects is essential.
>
> ### **Improving Generalization and Robustness**
> AI often fails to generalize as robustly as humans do [4,5] suffering from **shortcut learning**.
>
> Lukas Muttenthaler [1] highlights that whereas human conceptual knowledge is hierarchically organized from fine- to coarse-scale distinctions [6], model representations do not accurately capture all these levels of abstraction.
>
> They find that human-aligned models perform better on a diverse set of machine learning tasks, increasing generalization and out-of-distribution robustness.
>
> Thus, infusing neural networks with additional human knowledge yields a best-of-both-worlds representation that is both more consistent with human cognitive judgements and more practically useful, paving the way towards more robust, interpretable, and human-aligned artificial
> intelligence systems.
>
> ### **Enhancing Anthropomorphism and Interaction Quality**
>
> While the substrates differ, human-like object concept representations emerge naturally in multimodal large language models [3].
>
> Aligning AI with humans is not solely about imitative behavior. It improves the interpretability of AI outputs, making the reasoning processes and decisions more intuitive, transparent, and predictable for human users.
>
> Furthermore, Binz [2] demonstrates that a highly aligned model can serve as a powerful tool for understanding the human mind and aids in the discovery of cognitive theories.
>
> In summary, while we acknowledge the essential differences between AI and humans, human alignment is the bridge that translates computational power into robust, interpretable, and human-like intelligence.

---

> ### Author Response · Authors · 2025-11-24
>
> ## 5. Combining your response of (3), in my understanding, LLMs are used only to normalize the dimensions identified by Jackknife into certain fotmats (the answer in Fig. 2 and the assistant messages in Fig. 4), is this correct?
> ### Response:
> Yes, your understanding is correct. The LLM (we use Qwen-72B-Instruct) is primarily used to translate this identified **discrete dimension** (core cognitive dimension to each triplet) into a coherent natural language reasoning (the answer in Fig. 2).
>
> We present a specific example below.
> ***
> **Triplet objects**: prune, nail polish and diskette
>
> **Core cognitive dimension**: metallic/artificial.
>
> **The prompt to LLM (Qwen-72B-Instruct)** :
>
> You're a knowledgeable expert. Given three objects prune, nail polish and diskette. Here are their brief descriptions below.
>
> prune: dried plum
>
> nail polish: a cosmetic lacquer that dries quickly and that is applied to the nails to color them or make them shiny
>
> diskette: a small plastic magnetic disk enclosed in a stiff envelope with a radial slit; used to store data or programs for a microcomputer.
>
> With the above descriptions, you understand three objects prune, nail polish and diskette simply. Next, you need to incorporate the vast amount of knowledge you have to understand them further.
>
> You need to think carefully and analyse in terms of metallic or artificial. Next, you need to give your analysis.
>
> Here are some criteria you need to follow:
>
> 1.You only need to give your analysis. You must keep words in 200 words. You don't need to output the analysis process and how you analyse.
>
> 2.You need to analyse prune, nail polish and diskette based on your vast knowledge, known information and analytical perspective which is metallic or artificial.
>
> 3.Your analysis must be objective, reasonable, fair and logical.
>
> 4.If there is not much relationship between the angle of analysis, "metallic or artificial" and one or more of prune, nail polish and diskette, please do not carry out a forced analysis because forcing the analysis in terms of metallic or artificial is not permissible to ensure your analysis is reasonable.
>
> 5.You must strictly follow the below response template to output with your analysis.
>
> Here is your response template, You have to follow the template output given below.
>
> Analysis: {your analysis}
>
> **The output from LLM**:
>
> Analysis: Prune, being a dried fruit, is neither metallic nor artificial; it is a natural product derived from agriculture. Nail polish, on the other hand, is an artificial product composed of various chemicals, including solvents, resins, and pigments, designed for cosmetic use. Diskette, while not metallic in its entirety, contains metallic components within its magnetic coating, which is essential for data storage, but it is primarily a synthetic, artificial device used in technology.
> ***
> Subsequently, the output from LLM, triplet captions, and odd-one-out option are combined to construct training data for fine-tuning MLLMs (Qwen-2.5-7B-Instruct and Gemma3-12B-it).
>
> ## 6. Remember to explicitly write down "representational similarity matrix" before you first use "RSM".
> ### Response:
> We thank the reviewer for the careful reading. We have corrected in the revised manuscript (highlighted in orange).
>
>
>
> ### References:
> [1] Muttenthaler, L., Greff, K., Born, F. et al. Aligning machine and human visual representations across abstraction levels. Nature 647, 349–355 (2025). https://doi.org/10.1038/s41586-025-09631-6
>
> [2] Binz, M., Akata, E., Bethge, M. et al. A foundation model to predict and capture human cognition. Nature 644, 1002–1009 (2025). https://doi.org/10.1038/s41586-025-09215-4
>
> [3] Du, C., Fu, K., Wen, B. et al. Human-like object concept representations emerge naturally in multimodal large language models. Nat Mach Intell 7, 860–875 (2025). https://doi.org/10.1038/s42256-025-01049-z
>
> [4] Lake, B. M., Ullman, T. D., Tenenbaum, J. B. & Gershman, S. J. Building machines that learn and think like people. Behav. Brain Sci. 40, e253 (2017).
>
> [5] Geirhos, R. et al. Generalisation in humans and deep neural networks. In Proc. Advances in Neural Information Processing Systems Vol. 31 (eds Bengio, S. et al.) 7538–7550  (Curran Associates, 2018).
>
> [6] Hebart, M. N., Zheng, C. Y., Pereira, F. & Baker, C. I. Revealing the multidimensional  mental representations of natural objects underlying human similarity judgements.  Nat. Hum. Behav. 4, 1173–1185 (2020).

---

> > ### Comment · Reviewer_tnH7 · 2025-11-25
> >
> > Thanks for the clarification. I would like to raise my score.

---

> > > ### Author Response · Authors · 2025-11-26
> > > **Thanks for your recognition!**
> > >
> > > We would like to express our sincere gratitude to the reviewer for spending time and effort in providing constructive suggestions for our manuscript and for recognizing our work. We believe that under your guidance, our manuscript has achieved a higher level in terms of content readability and experimental integrity.

---

### Official Review · Reviewer_am6s · 2025-10-28

**Soundness:** 3
**Presentation:** 3
**Contribution:** 3
**Rating:** 6
**Confidence:** 3

**Summary:**

The paper proposes a training paradigm that goes past imitating human answers in triplet odd-one-out tasks. The authors first infer a core cognitive dimension presumed to underlie each O1O triplet, then inject that dimension—expressed in natural language—into instruction-tuning data for a vision-language model. They evaluate on held-out O1O prediction, representational similarity agreement with human behavior and fMRI, and broad multimodal benchmarks. The submission also compares against Centaur, a recent cognition-oriented foundation model.

**Strengths:**

Overall, the paper advances a clear and readily implementable idea: injecting interpretable cognitive dimensions into instruction tuning, which can be applied to off-the-shelf MLLMs (e.g., Qwen2.5-VL) while boosting scientific interpretability. Its evaluation strategy is genuinely multi-axis—combining O1O accuracy, behavior–RSM agreement, whole-brain RSA, and transfer to MMMU/MMMU-Pro—to probe behavioral, neural, and functional performance. Including Centaur as a comparator grounds the work within the emerging “models of cognition” literature and strengthens its timeliness. Moreover, the dimension-injection template appears broadly generalizable beyond object understanding to actions and scenes, naturally dovetailing with wider RSA frameworks for higher-level cognition.

**Weaknesses:**

1.Dimension inference — validation & stability: Create a human-labeled basis-dimension subset for sampled triplets; report agreement and κ with inferred labels; assess top-1 stability via bootstrap and alternative heuristics (top-k voting, ablation-magnitude thresholds).
2.Controls & fair comparisons: Add ablations to rule out prompt-length/attention effects (correct vs shuffled vs adversarial prompts; same-length null control; compare to plain rationales without dimensions). Ensure baseline parity by unifying zero-/few-shot settings (incl. Centaur), sweeping decoding parameters, and reporting mean ± sd across seeds/prompts.
3.Data hygiene & generalization: Provide explicit image-level and object-set splits and include an object-held-out evaluation where categories never appear in training; document leakage checks.
4.Reporting rigor (neuro + benchmarks): For RSA, report subject-level reliability/noise ceilings, apply FDR/permutation corrections, and run partial RSA controlling low-level and semantic confounds; avoid mapping PFC RSA directly to planning/decision. For MMMU-Pro, state whether vision-only was used and whether OCR prompts or CoT were enabled, and relate outcomes to known sensitivities.

**Questions:**

1.How accurate and stable is your core-dimension inference? Any human-rated rationale set you can release?
2.Are all baselines evaluated under matched shot count, prompts, and decoding choices? If not, please add a controlled comparison.
3.What safeguards ensure no train/test image or object leakage across O1O and RSA experiments?
4.How sensitive is your method to the choice of base model (e.g., Qwen2.5-VL vs other MLLMs) and to LoRA rank/where adapters are placed? Pointers to implementation details are welcome.

---

> ### Author Response · Authors · 2025-11-21
>
> We thank the am6s reviewer for recognizing and supporting our work. We have thoroughly revised the manuscript based on your feedback, with changes highlighted in green, and have submitted the updated version. We use red corrections in our manuscript for issues common to all reviewers. Below are our detailed responses to your comments.
>
> ## 1. How accurate and stable is your core-dimension inference? Any human-rated rationale set you can release?
> ### Response:
> (1) We believe our core dimension to each triplet is accurate and stable and you can trust it completely. The core-dimension to each triplet is one of 66 dimensions proposed by Martin Hebart [1,2]. And the 66 dimensions have been verified through large-scale human. We employ Jackknife method to pinpoint core dimension to each triplet among 66 dimensions. The identified dimension is demonstrably accurate and stable.
>
> (2) We will release all training sets on huggingface if our paper can be accepted, not just human-rated rationale set. These sets will greatly benefit the research community.
>
> (3) The 66 cognitive dimensions have been provided in Appendix A.6 Table 9 highlighted in red.
>
> ## 2. Are all baselines evaluated under matched shot count, prompts, and decoding choices? If not, please add a controlled comparison.
>
> ### Response:
> We evaluated under matched zero-shot, prompts, and decoding choices expect Centaur.
>
> Our training and evaluation sets are instruction datasets. It is unfair to evaluate Centaur utilizing these datasets because Centaur is a pretrained model after finetuning. It lacks instruction-following capability. Therefore, we follow the script (https://github.com/marcelbinz/Llama-3.1-Centaur-70B/blob/main/run_minimal.py) provided by authors of Centaur to evaluate accuracy on Odd-One-Out task. Table 1 below presents ablation experiments on Centaur in few-shot and zero-shot settings.
>
> Table 1. Odd-One-Out Accuracy with few-shot and zero-shot settings
> | N-Shot | Accuracy |
> |---------|-------|
> | Centau$_{0shot}$ | 9.312% |
> | Centaur$_{3 shot}$ | 26.95% |
> | Centaur$_{5 shot}$| 51.96% |
> | Gemma3-12B-it$_{0 shot}$ | 55.766% |
> | CogAligner$_{ours, Gemma3-12B-it, 0 shot}$ | **57.279%** |
> | Qwen-2.5-VL-7B-Instruct$_{0 shot}$ | 56.63% |
> | CogAligner$_{ours, Qwen-2.5-VL-7B-Instruct, 0 shot}$ | **59.29%** |
>
> ## 3. What safeguards ensure no train/test image or object leakage across O1O and RSA experiments?
> ### Response:
> We use multi-image training for the O1O task and single-image inference in RSA experiments. The O1O train and test data are disjoint, with no overlap between them. Moreover, there is no overlapping data between O1O and RSA experiments.
>
> Thus, we ensure no train/test image or object leakage across O1O and RSA experiments.
>
> ## 4. How sensitive is your method to the choice of base model (e.g., Qwen2.5-VL vs other MLLMs) and to LoRA rank/where adapters are placed? Pointers to implementation details are welcome.
> ### Response:
> Our data-driven method is not sensitive to the choice of base model and to LoRA rank.
>
> (1) We finetune two multimodal large models from different families and parameters, Qwen-2.5-VL-7B-Instruct and Gemma3-12B-it. Table 2 shows the accuracy of them and corresponding baseline (Chain-of-Thought, COT) on O1O task validation set below.
>
> Table 2. O1O Accuracy across baseline and finetuned models
> | Model | Accuracy |
> |---------|-------|
> | Gemma3-12B-it | 55.766% |
> | CogAligner$_{ours, Gemma3-12B-it}$ | **57.279%** |
> | Qwen-2.5-VL-7B-Instruct | 56.63% |
> | CogAligner$_{ours, Qwen-2.5-VL-7B-Instruct}$ | **59.29%** |
>
> We get same expected results and CogAligners outperform baseline models. These results indicate that our method is not sensitive to the choice of base model. We will use more base models from different model families and parameters to verify our method, and release results on the openreview after experiments finish.
>
> (2) In our experiment, we adopt the recommended LoRA configuration (e.g., rank=16, alpha=32) and insert them into linear layers (language model) and projection layers with freezing vision model.
>
> These settings are selected based on extensive preliminary studies and align with best practices for LoRA finetuning of MLLMs. We will include full implementation details in Appendix A.3 Table 4 of the revised manuscript highlighted in green. If our paper can be accepted, we will release all source codes on github.
>
> ### References:
> [1] Hebart, M. N., Zheng, C. Y., Pereira, F. & Baker, C. I. Revealing the multidimensional mental representations of natural objects underlying human similarity judgements. Nat. human behaviour 4, 1173–1185 (2020).
>
> [2] Hebart, M. N. et al. THINGS-data, a multimodal collection of large-scale datasets for investigating object representations in human brain and behavior. Elife 12, e82580 (2023).

---

> ### Author Response · Authors · 2025-11-28
> **Thank you once again!**
>
> Thank you once again for your constructive feedback on our submission.
>
> We submitted our rebuttal, where we addressed your main concerns.
>
> We hope that our responses, including the additional experiments and clarifications, have fully addressed your initial questions and concerns.
>
> If, after reviewing our rebuttal, you have any further lingering questions or require any additional discussion points, we would be very happy to engage. Please do not hesitate to let us know.
>
> Thank you for your time and expertise in reviewing our work.

---

### Official Review · Reviewer_LW1V · 2025-10-28

**Soundness:** 3
**Presentation:** 3
**Contribution:** 2
**Rating:** 4
**Confidence:** 2

**Summary:**

Multimodal large language models now deliver near-human performance on diverse multimodal reasoning tasks, despite being trained with next-token prediction. A key open question is whether they form human-like visual concepts and exhibit human-like behavior. Using the Odd-One-Out task, their predictions poorly matched human choices. The authors addressed this by extracting core cognitive dimensions and judgment bases from human O1O behavior, then fine-tuning Qwen2.5-VL with these dimensions. This yielded markedly better alignment with human behavior while preserving—and sometimes improving—downstream task performance. Searchlight RSA and cortical projection analyses showed increased activation in regions linked to planning and decision-making, such as the prefrontal cortex, in the fine-tuned model, offering a potential neuroscientific account of the gains and human-like alignment.

**Strengths:**

-   The paper is well-structured; the method is reasonable and supported by comprehensive evidence and references.
-   Through careful data collection and fine-tuning, this work effectively improves the alignment between MLLMs' cognitive processes and those of humans.
-   The experimental design is sound.

**Weaknesses:**

-   Based on the results in Table 1, the method seems to offer limited improvements on general benchmarks.
-   The authors stress addressing the critical but underexplored challenge of bridging the gap between human-level performance and human-like cognitive processes in MLLMs. However, this may not actually be a limitation of MLLMs. In other words, why do MLLMs need human-like cognitive processes, given their learning mechanisms are quite different from humans?

**Questions:**

Please kindly find it in the weaknesses section.

---

> ### Author Response · Authors · 2025-11-21
>
> We sincerely appreciate your recognition of the novelty of our work. Thank you for taking the time to point out our weaknesses. We have thoroughly revised the manuscript based on your feedback, with changes highlighted in blue, and have submitted the updated version. We use red corrections in our manuscript for issues common to all reviewers. Below are our detailed responses to your comments.
>
> ## 1. Based on the results in Table 1, the method seems to offer limited improvements on general benchmarks.
> ### Response:
> Our primary objective is not to enhance general model capabilities, but rather to align the MLLMs with core human cognitive dimensions via fine-tuning to achieve human-like behavior. Consequently, we utilize Odd-One-Out behavioral data instead of traditional datasets typically used for boosting general performance.
> However, we are surprised to observe that a model more closely aligns with human behavior does not suffer from alignment tax or performance degradation. On the contrary, it not only preserves its original capabilities but also achieves slight improvements on general benchmarks.

---

> ### Author Response · Authors · 2025-11-21
>
> ## 2. Why do MLLMs need human-like cognitive processes, given their learning mechanisms are quite different from humans?
> ### Response:
> We appreciate the reviewer's insightful question regarding the necessity of human-like cognitive processes in MLLMs given their distinct learning mechanisms. While it is true that MLLMs and humans learn differently, evidence from extensive prior research [1-6] demonstrates that alignment with human cognitive dimensions is not merely a philosophical goal but a functional necessity for generalization and interpretability.
>
> (1) **Generalization**: While MLLMs learn differently from humans, previous work [1,3,4,5] demonstrate that aligning with human cognitive dimensions improves general capabilities.
>
> The research of Du [6] observed that human-like object concept representations emerge naturally in high-performing models, suggesting that these dimensions (e.g., animal-related, metallic, food-related) are efficient schemas for organizing information about the natural world.
>
> In the current work, large models trained solely on next-token prediction often fail to capture the causal "why" behind a decision, leading to limited predictive accuracy in behavioral tasks. However, when we explicitly fine-tuned the model (CogAligner) using these core cognitive dimensions, we observed a "favorable trade-off" where the model not only aligned better with humans but also achieved performance improvements on general downstream benchmarks like MMMU and MMMU-Pro.
>
> This indicates that human-like cognitive dimensions provide a structured reasoning framework that enhances a model's ability to generalize to complex, multi-discipline tasks, preventing it from relying on superficial statistical correlations.
>
> (2) **Interpretability**: Interpretable and trustworthy reasoning for MLLMs to be integrated into human workflows, their decision-making process must be intelligible to users.
>
> Variational Interpretable Concept Embeddings work emphasizes that deriving embeddings from human behavior yields dimensions that are inherently interpretable (e.g., functional or structural properties) rather than opaque mathematical vectors. Sundaram [5] corroborates that aligning to human perceptual judgments creates representations that are beneficial for retrieval-augmented generation (RAG) and dense prediction tasks, effectively injecting an inductive bias about human perceptual knowledge.
> Current alignment techniques (like RLHF) focus on mimicking the outcome (behavioral alignment). However, mimicking the output without the underlying process is insufficient for complex reasoning. Simply matching the outcome of a human decision does not guarantee an alignment with the cognitive processes that led to it. By leveraging core cognitive dimensions (the "why"), we move beyond superficial mimicry.
>
> Although MLLMs use different learning mechanisms (backpropagation vs. biological plasticity), they must operate in a human-centric world. The "limitation" is not the difference in mechanism, but the divergence in representation. Aligning these representations does not compromise performance but rather enhances it across diverse tasks [1,4].
>
> Therefore, human-like cognitive processes are a necessary target to ensure models are robust, generalizable, and interpretable partners for human users.
>
> ### References:
> [1] Muttenthaler, L., Greff, K., Born, F. et al. Aligning machine and human visual representations across abstraction levels. Nature 647, 349–355 (2025). https://doi.org/10.1038/s41586-025-09631-6
>
> [2] Muttenthaler, L., Zheng, C., McClure, P., Vandermeulen, R., Hebart, M., & Pereira, F. (2022). VICE: Variational Interpretable Concept Embeddings. In Advances in Neural Information Processing Systems (pp. 33661–33675). Curran Associates, Inc..
>
> [3] Muttenthaler, L., Dippel, J., Linhardt, L., Vandermeulen, R., & Kornblith, S. (2023). Human alignment of neural network representations. In 11th International Conference on Learning Representations, ICLR 2023, Kigali, Rwanda, Mai 01-05, 2023. OpenReview.net.
>
> [4] Fu, S., Tamir, N., Sundaram, S., Chai, L., Zhang, R., Dekel, T., & Isola, P. (2023). DreamSim: Learning New Dimensions of Human Visual Similarity using Synthetic Data. In Advances in Neural Information Processing Systems (pp. 50742–50768). Curran Associates, Inc..
>
> [5] Sundaram, S., Fu, S., Muttenthaler, L., Tamir, N., Chai, L., Kornblith, S., Darrell, T., & Isola, P. (2024). When does perceptual alignment benefit vision representations?. In Advances in Neural Information Processing Systems (pp. 55314–55341). Curran Associates, Inc..
>
> [6] Du, C., Fu, K., Wen, B. et al. Human-like object concept representations emerge naturally in multimodal large language models. Nat Mach Intell 7, 860–875 (2025). https://doi.org/10.1038/s42256-025-01049-z

---

> > ### Comment · Reviewer_LW1V · 2025-11-25
> >
> > Thank you for your reply. It has addressed some of my concerns. I agree with the importance of generalization and interpretability, but I still cannot fully agree that machines need human-like cognitive processes. I am not an expert in this field. Overall, I will keep my score (4) and confidence (2).

---

> > > ### Author Response · Authors · 2025-11-25
> > >
> > > Thank you for taking the time to review our rebuttal. We are pleased that your concerns have been addressed.

---

### Official Review · Reviewer_xWtF · 2025-10-29

**Soundness:** 2
**Presentation:** 3
**Contribution:** 2
**Rating:** 4
**Confidence:** 4

**Summary:**

The authors propose a method that first extracts core cognitive dimensions from human experimental data, and then uses these dimensions to guide model fine-tuning. They demonstrate that this approach enhances the model’s similarity to humans at both behavioral and neural levels, and provides a new perspective for developing more interpretable and cognitively aligned artificial intelligence systems.

**Strengths:**

The paper presents a clear research motivation, a well-organized structure, and well-designed figures that effectively illustrate its methodology. The authors employ SPoSE and Jackknife methods to extract latent cognitive dimensions from human Odd-One-Out experimental data, and use these dimensions to fine-tune a multimodal large language model. Additionally, through prompt engineering, the model’s training is aligned with human neural representations. Finally, neural data are used to validate the effectiveness of this cognitive alignment, and the results show that the model exhibits higher consistency with humans at both the behavioral and neural levels.

**Weaknesses:**

1.	Although the paper is well-structured and logically coherent, its novelty is relatively limited. The proposed Cognitive Dimension Alignment is conceptually very similar to existing Concept Bottleneck Models and subsequent concept-based interpretability studies. Several recent works on concept-based LLMs/VLMs have already achieved model alignment through intermediate semantic layers or interpretable latent dimensions. The paper’s so-called core cognitive dimensions (e.g., metallic, food-related, animal-related) are closer to interpretable semantic concepts rather than true cognitive processes. These dimensions mainly capture shared perceptual or categorical attributes in human semantics but fail to reflect higher-order cognitive abilities such as abstraction, causal reasoning, emotion, or behavioral tendencies. Therefore, defining these semantic features directly as “cognitive dimensions” is theoretically unconvincing. To strengthen the argument of cognitive alignment, the authors should further clarify how these dimensions relate to cognitive dimensions in psychology or provide empirical evidence showing that the model’s decision-making process truly aligns with human cognitive structures.

2.	The evidence for neural alignment remains insufficient. The NSD dataset includes only eight participants, resulting in a limited sample size. Although the RSA results are suggestive, the paper lacks rigorous statistical validation. The authors are encouraged to perform more systematic significance testing and cross-subject consistency analyses to enhance the robustness and credibility of their findings.

3.	The current experiments are conducted solely on the Qwen2.5-VL model. Verifying whether the proposed method generalizes to other architectures (e.g., GPT-4 or Gemini) would greatly strengthen the claims. In fact, based on the presented tasks and figures, preliminary observations suggest that under simple prompt conditions, models like GPT-4 or GPT-5 can already correctly identify cognitive dimensions and make human-consistent choices. Therefore, broader comparative experiments across multiple models and settings are necessary to establish the unique contribution of this work.

4.	Conducting fine-tuning experiments with different data ratios is a common practice and cannot be considered a major innovation. The paper should include a more critical discussion of the limitations and potential biases of the proposed method, rather than emphasizing only the positive results. Additionally, incorporating human behavioral or participant-based analyses could further validate the proposed model’s alignment with human cognition.

**Questions:**

1.	On the Definition of “Cognitive Dimensions”
The paper defines features such as metallic, food-related, and animal-related as “cognitive dimensions.” However, these dimensions appear to be closer to semantic concepts or interpretable features. Could the authors clarify, from both theoretical and empirical perspectives, how these dimensions differ from traditional concept representations?

2.	Statistical Robustness and Variability
It is recommended that the authors report the variance or confidence intervals of key experimental metrics to assess sensitivity to random initialization, data splits, or sample size. This analysis would help improve the credibility and stability of the conclusions.

3.	Interpretability Validation of Cognitive Dimensions
The authors are encouraged to include more visualizations or ablation studies to demonstrate whether the SPoSE + Jackknife–derived dimensions correspond to human-understandable concepts. Without such validation, the extracted “dimensions” might be statistically driven embeddings rather than genuinely cognitive representations.

---

> ### Author Response · Authors · 2025-11-21
>
> We thank the xWtF reviewer for recognizing and supporting our work. We appreciate the opportunity to explain the definition of "Cognitive Dimensions". Below are our detailed responses to your comments. The common issues are highlighted in red.
>
> ## 1. Why our "Cognitive Dimensions" differ from traditional semantic concepts or interpretable features?
> ### Response:
> We understand reviewer concern that the dimensions (e.g., metallic, food-related) are seen as "interpretable semantic concepts" rather than true "cognitive processes" and the approach is likened to Concept Bottleneck Models.
>
> We respectfully disagree with the characterization of these features as merely "semantic concepts" or simple attributes. Our definition of "Cognitive Dimensions" is strictly derived from 66 dimensions proposed by Martin Hebart [1], not just static object properties.
>
> (1) Distinction from Semantic Features: As explained by Martin Hebart [1], traditional semantic features are often binary and linguistically derived (e.g., "is a dog"). In contrast, the cognitive dimensions corresponding to 66 dimensions used in our work are continuous, non-negative dimensions derived via SPoSE (Sparse Positive Object Similarity Embedding) from 4.7 million behavioral triplets. These dimensions represent the specific axes along which humans compute similarity to make decisions. They are not just "what" the object is, but "how" humans weigh features to reach a decision.
>
> (2) Novelty over Concept Bottleneck Models: Unlike Concept Bottleneck Models that rely on pre-defined, annotated concepts, our dimensions are discovered from behavior. These 66 discovered dimensions have been verified by large-scale human from Martin Hebart's experiments [1,6]. Therefore, we can’t simply equate semantic concepts with cognitive dimensions.
>
> ## 2. The evidence for neural alignment remains insufficient. The NSD dataset includes only eight participants, resulting in a limited sample size. Although the RSA results are suggestive, the paper lacks rigorous statistical validation.
> ### Response:
> While the eight subjects are a relatively small sample, the NSD dataset is widely regarded as one of the highest-quality and most extensive functional magnetic resonance imaging (fMRI) datasets currently available for this type of research. Furthermore, our RSA method is recognized within the field [2-4]. If even larger and higher quality 7T data becomes available in the future, we will validate our method on these data.
>
> In each subject RSA experiment, the means of CogAligner and Qwen-2.5-7B-Instruct results are very close and the means eliminate the differences between them. So, we have performed the two-sample Kolmogorov-Smirnov test for goodness of fit about RSA experiments in Fig.8.
>
> This test compares the underlying continuous distributions F(x) and G(x) of two independent samples. Suppose we wish to test the null hypothesis that two samples were drawn from the same distribution. We choose a confidence level of 95%; that is, we will reject the null hypothesis in favor of the alternative if the p-value is less than 0.05. The results of two-sample Kolmogorov-Smirnov test are shown in Table 1 below.
>
> Table 1. Two-sample Kolmogorov-Smirnov Test Result
> | Subject | KS |  P | Significant Difference |
> | :---------: | :--------: | :-------------: | :--------: |
> | 1 | 0.0095 | < 0.05| Yes |
> | 2 | 0.0151 | < 0.05 | Yes |
> | 5 | 0.0151 | < 0.05 | Yes |
> | 7 | 0.0149 | < 0.05 | Yes |
>
> The results of the hypothesis test indicate that the difference is significant to each subject contrast result. We have added the statistical validation in description of Fig.8 highlighted in red.
>
> ## 3. The current experiments are conducted solely on the Qwen2.5-VL model. Verifying whether the proposed method generalizes to other architectures (e.g., GPT-4 or Gemini) would greatly strengthen the claims. Broader comparative experiments across multiple models and settings are necessary to establish the unique contribution of this work.
> ### Response:
> Due to tight rebuttal timeline, we conduct supplementary experiments to test additional models accuracy on Odd-One-Out task, including GPT-3.5 and Gemini Pro vision 1.0, and Gemma3-12B-it (extended verification of our method). Detailed accuracy results for Odd-One-Out task are shown in Table 2 below.
>
> Table 2. Odd-One-Out Accuracy on Different Models
> | Model | Accuracy | Noise Ceiling |
> |---------|-------|-------|
> | GPT-3.5 | 56.75% | 65.1% |
> | Gemini Pro Vision 1.0 | 63.4% | 73.8% |
> | Gemma3-12B-it | 55.766% | 68.74% |
> | CogAligner$_{Gemma3-12B-it}$ | 57.279% | 68.74% |
>
> The verification results for Gemma3-12B-it have been added to Table 1&2 (highlighted in red) in the revised manuscript.

---

> ### Author Response · Authors · 2025-11-21
>
> ## 4. Statistical Robustness and Variability It is recommended that the authors report the variance or confidence intervals of key experimental metrics to assess sensitivity to random initialization, data splits, or sample size. This analysis would help improve the credibility and stability of the conclusions.
> ### Response:
> (1) We provide 95% confidence intervals for human consistency in Fig.6 and show them in Table 3 below.
>
> Table 3. 95% Confidence Intervals for Human Consistency
> | Model | Human Consistency | lower bounds | upper bounds |
> |----------------|-------|--------|----------|
> | Human | 91.05% | 89.94% | 92.04% |
> | Centaur$_{3 shot}$ | 48.39% | 43.14% | 53.39% |
> | Centaur$_{5 shot}$ | 65.13% | 61.21% | 68.80% |
> | Qwen-2.5-VL-7B-Instruct (COT) | 73.30% | 69.51% | 76.80% |
> | BehavImitator | 80.22% | 77.12% | 83.04% |
> | CogAligner (ours) | **80.93%** | 77.86% | 83.73% |
>
> (b) We also have performed the two-sample Kolmogorov-Smirnov test and added results in Fig.8 (See Table 1 of Response 2).
>
> In the updated manuscript, we highlight these statistical tests to Fig.6 and Fig.8 in red.
>
> ## 5. Interpretability Validation of Cognitive Dimensions The authors are encouraged to include more visualizations or ablation studies to demonstrate whether the SPoSE + Jackknife–derived dimensions correspond to human-understandable concepts. Without such validation, the extracted “dimensions” might be statistically driven embeddings rather than genuinely cognitive representations.
> ## Response:
> Our cognitive dimensions are based on 66 dimensions proposed by Martin Hebart [1,6]. And these 66 dimensions have been verified by large-scale human explained in Martin Hebart’s experiment. Therefore, we don’t need to verify again. Next, we employ Jackknife method to derive each triplet’s core-cognitive dimension. All processes are not statistically driven embeddings rather than genuinely cognitive representations. We have added our 66 cognitive dimensions in Appendix A.6 Table 9.
>
> ### References:
> [1] Hebart, M. N., Zheng, C. Y., Pereira, F. & Baker, C. I. Revealing the multidimensional mental representations of natural objects underlying human similarity judgements. Nat. human behaviour 4, 1173–1185 (2020).
>
> [2] Wang, A.Y., Kay, K., Naselaris, T. et al. Better models of human high-level visual cortex emerge from natural language supervision with a large and diverse dataset. Nat Mach Intell 5, 1415–1426 (2023). https://doi.org/10.1038/s42256-023-00753-y
>
> [3] Du, C., Fu, K., Wen, B. et al. Human-like object concept representations emerge naturally in multimodal large language models. Nat Mach Intell 7, 860–875 (2025). https://doi.org/10.1038/s42256-025-01049-z
>
> [4] Tomoyasu Horikawa ,Mind captioning: Evolving descriptive text of mental content from human brain activity.Sci.Adv.11,eadw1464(2025).DOI:10.1126/sciadv.adw1464
>
> [5] Hodges, J.L. Jr., "The Significance Probability of the Smirnov Two-Sample Test," Arkiv fiur Matematik, 3, No. 43 (1958), 469-486.
>
> [6] Hebart, M. N. et al. THINGS-data, a multimodal collection of large-scale datasets for investigating object representations in human brain and behavior. Elife 12, e82580 (2023).

---

> ### Author Response · Authors · 2025-11-28
> **Thank you once again!**
>
> Thank you once again for your constructive feedback on our submission.
>
> We submitted our rebuttal, where we addressed your main concerns.
>
> We hope that our responses, including the additional experiments and clarifications, have fully addressed your initial questions and concerns.
>
> If, after reviewing our rebuttal, you have any further lingering questions or require any additional discussion points, we would be very happy to engage. Please do not hesitate to let us know.
>
> Thank you for your time and expertise in reviewing our work.

---

### Author Response · Authors · 2025-11-22

Dear reviewers,

Thank you for your valuable time, insightful comments, and constructive suggestions.

We have thoroughly revised the manuscript based on the feedback received, and the latest submission reflects these changes. To ensure clarity for the reviewers, we have highlighted all revisions addressing each reviewer's comments in different colors.

Common Issues: $\color{red}{Red}$, Reviewer $\color{blue}{LW1V}$: $\color{blue}{Blue}$, Reviewer $\color{orange}{tnH7}$: $\color{orange}{Orange}$, and Reviewer $\color{green}{am6s}$: $\color{green}{Green}$.

Our detailed, point-by-point responses to each reviewer's comments have been individually added in the corresponding chat boxes. We are confident that the manuscript has been substantially improved in both clarity and experimental rigor thanks to your insightful guidance.

---

### Author Response · Authors · 2025-12-03
**Summary of Rebuttal (1/2)**

We have summarized the reviewers' comments (strengths and weaknesses) and our corresponding response during rebuttal period.

## **Strengths:**

### 1. **Clear Motivation**

We present clear and well-structured argument for moving beyond simple behavioral mimicry to aligning with underlying core cognitive dimensions.

### 2. **Reasonable Method**

The approach of extracting latent core cognitive dimensions using SPoSE and Jackknife from the THINGS dataset is considered reasonable and innovative.

### 3. **Comprehensive Evaluation**

Reviewers appreciate the multi-axis evaluation strategy, which covers behavioral accuracy (Odd-One-Out task), general benchmarks (MMMU and MMMU-Pro), and searchlight representational similarity analysis (RSA) and cortical projection analyses—to probe behavioral, neural, and functional performance.

### 4. **Strong Baselines**

Including **Centaur** [1] as a comparator grounds the work within the emerging "models of cognition" literature and strengthens its timeliness.

### 5. **Extending Method**

Reviewers appreciate the dimension-injection appears broadly generalizable beyond object understanding to actions and scenes.

## **Issues:**

### 1. **Method Generalization**
Reviewers (xWtF, am6s, and tnH7): the method is verified only on the Qwen-2.5-VL-7B. Reviewers question whether the method generalizes to other model families or sizes.

Our response: we apply our method to a completely different model family, Gemma3-12B-it. Our finetuned model **Cogaligner$_{Gemma3-12B-it}$** improves behavioral accuracy (Odd-One-Out task) from 55.766% (Gemma3-12B-it baseline) to **57.279%**. Furthermore, Cogaligner$_{Gemma3-12B-it}$ maintains or improves general performance on MMMU and MMMU-Pro datasets. These representative results demonstrate that our method is model-agnostic and scalable.

### 2. **Definition of Cognitive Dimensions**
Reviewer (xWtF): the **cognitive dimensions** (e.g., food-related) seems like traditional concept representations.

Our response: our definition of "cognitive dimensions" is strictly derived from **66 dimensions** proposed by **Martin Hebart** [2], not just static object properties. The cognitive dimensions corresponding to 66 dimensions used in our work are continuous, non-negative dimensions derived via SPoSE (Sparse Positive Object Similarity Embedding) from 4.7 million behavioral triplets. These dimensions represent the specific axes along which humans compute similarity to make decisions. They are not just "what" the object is, but "how" humans weigh features to reach a decision. These 66 discovered dimensions have been verified by large-scale human from Martin Hebart's experiments [2,3].
***
Reviewers (LW1V and tnH7): why align through human core cognitive dimensions, given machine and human different learning mechanisms.

Our response: prior work [1,4,5] indicates that, although the underlying mechanisms differ, advanced AI and human cognition can converge functionally. Focusing on aligning these functional aspects is essential. We provide detailed explanations from the perspective of **generalization, robustness,  and enhancing anthropomorphism and interaction quality**.

### 3. **Statistical Evidence**
Reviewers (xWtF and am6s): the **human consistency results** and **searchlight RSA** results lack statistical significance testing or confidence intervals.

Our response: we compute **95% confidence intervals** using 1000 bootstrap resamples for **human consistency** and performe **two-sample Kolmogorov-Smirnov test**  for **searchlight RSA**. All these results are significant.

### 4. **Benchmark and Fairness**
Reviewers (LW1V and tnH7): limited improvement and lower scores than leadboard published on general benchmarks (MMMU and MMMU-Pro).

Reviewers (am6s and tnH7): the fairness of comparing **CogAligner (zero-shot)** vs. **Centaur (few-shot)**.

Our response:

(1) **score discrepancy**: the higher scores reported on leaderboards are likely to utilize internal, highly optimized proprietary prompts, different system instructions or settings that are not fully public. We leverage open-source **lmms-eval** with default settings to retest for comparing fairly but obtaine lower results.

(2) **fairness**: Centaur (based on LLaMA-3.1-70B pretrained model) requires a few-shot setting to output valid formats. CogAligner (ours) operates in a zero-shot setting, which is generally considered more challenging than few-shot. CogAligner demonstrates superior alignment and interpretability, strengthening our claims.

(3) **trade-off**: our primary objective is not to enhance general model capabilities, but rather to align the MLLMs with core human cognitive dimensions via fine-tuning to achieve human-like behavior. Maintaining or even slightly improving performance on general benchmarks already constitutes a **favorable trade-off**.

---

> ### Author Response · Authors · 2025-12-03
> **Summary of Rebuttal (2/2)**
>
> ### 5. **Reproducibility and Details**
> Reviews (am6s and tnH7): 66 core cognitive dimensions, data leakage, undefined terminology, and typos.
>
> Our response:
>
> (1) **66 cognitive dimensions**: these cognitive dimensions have been added to Appendix A.6.
>
> (2) **data leakage**: training data and test data are strictly disjoint, ensuring no data leakage.
>
> (3) **undefined terminology and typos**: we revise these errors pointed out by reviewers and update in manuscript.
>
> ## **Conclusion:**
>
> We responses to reviewers one by one and **mainly discuss with reviewer tnH7**.
>
> During discussion period, **review tnH7 concerns are all have been addressed and get recognition and raising score**.
>
> ## **References:**
> [1] Binz, M., Akata, E., Bethge, M. et al. A foundation model to predict and capture human cognition. Nature 644, 1002–1009 (2025). https://doi.org/10.1038/s41586-025-09215-4
>
> [2] Hebart, M. N., Zheng, C. Y., Pereira, F. & Baker, C. I. Revealing the multidimensional mental representations of natural objects underlying human similarity judgements. Nat. human behaviour 4, 1173–1185 (2020).
>
> [3] Hebart, M. N. et al. THINGS-data, a multimodal collection of large-scale datasets for investigating object representations in human brain and behavior. Elife 12, e82580 (2023).
>
> [4] Muttenthaler, L., Greff, K., Born, F. et al. Aligning machine and human visual representations across abstraction levels. Nature 647, 349–355 (2025). https://doi.org/10.1038/s41586-025-09631-6
>
> [5] Du, C., Fu, K., Wen, B. et al. Human-like object concept representations emerge naturally in multimodal large language models. Nat Mach Intell 7, 860–875 (2025). https://doi.org/10.1038/s42256-025-01049-z

---

### Note · Authors · 2026-01-29

I have read and agree with the venue's withdrawal policy on behalf of myself and my co-authors.

---

### Meta-Review · Area_Chair_Focr · 2026-01-05

**Summary:**

Reviewers had several major concerns about novelty, statistical significance, generalization, and clarity, as well as minor concerns like listing limitations and performing certain ablations.

**Reviewer Concerns:**

While the authors put in a substantial amount of effort during the rebuttal period to try to address reviewer concerns (and largely addressed the concerns around clarity), I believe they have not fully addressed some of the major concerns:
- Statistical significance: only some stat sig tests were added, but other key results are still missing these and appear unlikely to be significant (e.g. Fig 5)
- Generalization to other models: authors added an experiment with a Gemma model, but the results are unconvincing (e.g., marginal improvement in O1O score, which reviewers also mentioned as a concern for the original Qwen experiments). Moreover, authors did not perform baseline prompt-based experiments with SOTA large-scale models as requested by some reviewers.

Some minor concerns were also not fully addressed:
- E.g., Data ratio ablations: Multiple reviewers mentioned this, but the authors did not run these ablations. Author arguments for why these ablations are unnecessary seemed unconvincing given multiple reviewers were concerned about it.

**Reviewer Scores:**

I believe only Reviewer tnH7 would have raised their score, possibly to around a score of 4 (originally 2).

---

### Decision · Program_Chairs · 2026-01-26

Reject